# Target-Driven Policy Optimization for Sequential Counterfactual Outcome Control

## Abstract

Identifying optimal intervention sequences from offline data to guide temporal systems toward target outcomes is a critical challenge with profound implications for fields like personalized medicine. While existing methods are mostly evaluated in offline settings, practical applications demand online, adaptive strategies that can respond in real-time. To address this, we propose **G**oal-conditioned **I**ntervention via **F**actual-**T**argeted Training (**GIFT**), a novel framework for learning sequential intervention policies from observational data. GIFT learns a goal-conditioned policy by rescaling rewards with clipped importance weights, stabilizing learning and steering toward the target. Under standard assumptions, the induced operator has a unique fixed point and our procedure converges to it. We also bound the bias from clipping and approximation via the gap to the policy's true value. Experiments show GIFT significantly outperforms existing methods in creating goal-conditioned policies for online deployment.

## 1 Introduction

A significant challenge in personalized medicine is to leverage observational data, such as Electronic Health Records (EHRs), to devise effective intervention strategies that steer a patient's physiological state towards a desired target (Figure 1a). This necessitates a shift from merely predicting counterfactual outcomes to proactively planning a sequence of interventions to achieve the goal.

Recent research (Wang et al., 2025) has formalized this as Sequential Counterfactual Target Achievement (SCTA), solved using counterfactual estimation (Lim et al., 2018; Bica et al., 2020; Melnychuk et al., 2022; Wang et al., 2024) or maximizing target achievement likelihood (Wang et al., 2025). These approaches rely on offline planning, where optimal intervention sequences are pre-computed via complex optimization (Figure 1b). However, this paradigm lacks real-time adaptability and suffers from high inference costs.

To address offline planning's limitations in adaptivity and efficiency, this study shifts from fixed intervention sequences to learning a real-time reactive policy. The problem is formulated as a goal-conditioned Markov Decision Process (MDP), with state defined by historical trajectory and goal, actions as interventions, and rewards tied to goal attainment (Figure 1c). The objective is to learn an optimal policy from observational data that efficiently guides the system to its target [1].

However, learning a policy for online deployment from fixed offline observational data encounters two key obstacles, which are distributional shift and limited success signals in the data (Levine et al., 2020). First is distributional shift, which arises when the learned policy explores state-action spaces not covered by the dataset, leading to value estimation errors and training instability (Fujimoto et al., 2019; Kumar et al., 2020). Second is sparse rewards, where success signals for goal achievement appear infrequently in

---

[1]"Target" and "goal" may be used interchangeably when unambiguous; in clinical contexts we prefer "target".

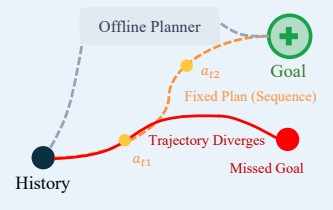 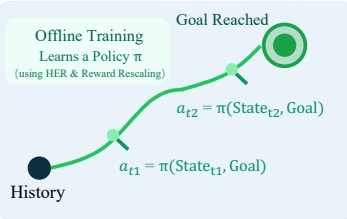

(a) Target achievement      (b) Previous offline planning      (c) GIFT: online policy

Figure 1: **Conceptual comparison of intervention paradigms. (a)** The problem of finding intervention sequences to guide a system toward a desired target. **(b)** The previous offline planning paradigm pre-computes a fixed action sequence. This static plan may fail as the trajectory diverges. **(c)** Our Approach, GIFT, learns a goal-conditioned online policy that dynamically adjusts actions to successfully reach the target.

the observational data, making it difficult for the model to discover useful behavioral patterns and thereby severely hindering the training process Sutton et al. (1998).

To address the aforementioned challenges, we propose the Goal-conditioned Intervention via Factual-Targeted Training (GIFT) approach, a novel framework specifically designed for learning goal-conditioned intervention strategies from observational data. GIFT addresses key issues through two core mechanisms. To tackle the sparse reward problem, GIFT incorporates the Hindsight Experience Replay (HER) mechanism (Andrychowicz et al., 2017), which relabels actually achieved future states in trajectories as virtual goals, transforming failed experiences into successful learning samples and alleviating data sparsity. To address the distribution shift problem, GIFT designs a reward rescaling mechanism that dynamically adjusts reward signals through bounded, clipped importance sampling weights, guiding value learning toward the evaluation policy distribution while suppressing the high variance issues of traditional methods, achieving a balance between bias and variance and stabilizing the offline training process. The main contributions of this work are summarized as follows:

- To the best of our knowledge, this is the first formulation of SCTA as a goal-conditioned MDP, highlighting its real-world significance.

- We propose the GIFT framework, which combines HER and reward rescaling mechanisms to learn goal-conditioned policies from offline data efficiently and stably, with theoretical guarantees including convergence proof and performance gap upper bound analysis.

- Experiments on synthetic and semi-synthetic datasets show GIFT significantly outperforms existing baselines in effectiveness, generalization, and efficiency.

## 2 RELATED WORK

**Offline Planning vs. Goal-Conditioned Policies.** The dominant paradigm for this problem is a two-stage "learn-then-plan" approach (Wang et al., 2025). First, a world model is trained on observational data to predict outcomes under hypothetical interventions. This line of work evolved from classical statistical methods (Robins, 1986; Robins et al., 2000; Fitzmaurice et al., 2008; Mortimer et al., 2005) to modern deep learning models for counterfactual estimation (Lim et al., 2018; Bica et al., 2020; Melnychuk et al., 2022; Wang et al., 2024) and likelihood maximization (Wang et al., 2025). A key innovation in this area was the use of the Transformer architecture (Vaswani et al., 2017) to better capture long-range dependencies (Hochreiter et al., 2001). However, all methods under this paradigm share a fundamental limitation: at inference time, they

require a costly optimization search to find a static, open-loop plan. Such plans are brittle to environmental stochasticity and ill-suited for real-time adaptation.

We frame the problem using goal-conditioned MDP to learn a reactive policy $\pi(a \mid s, g)$. Unlike typical robotics applications which use online simulators and methods like Hindsight Experience Replay (HER) (Andrychowicz et al., 2017), our work learns from a fixed, offline dataset. Adapting online methods to static data is challenging due to distributional shift (Levine et al., 2020) and sparse rewards. To our knowledge, we are the first to formulate SCTA as an offline, goal-conditioned RL task that learns an adaptive policy from such data. Compared with general goal-conditioned RL in robotics and vision, which often assumes unconfounded data and focuses on success probability or shortest-path style objectives (Eysenbach et al., 2020; Akella et al., 2023; Park et al., 2023; Zheng et al., 2024), our setting focuses on SCTA on observational cohorts where causal identifiability and offline distribution shift are central. GIFT addresses these issues by combining goal conditioning with HER and clipped-importance reward rescaling, yielding a modified soft Bellman operator with contraction and explicit bias bounds.

A comprehensive review, comparing offline planning and goal-conditioned policy paradigms and discussing distinctions from Dynamic Treatment Regimes (Murphy, 2003), is provided in the Appendix.

## 3 PROBLEM FORMULATION

We assume access to longitudinal observational dataset $\mathcal{D}$ comprising records from $N$ subjects: $\mathcal{D} = \left\{ \{\mathbf{X}_t^{(i)}, \mathbf{A}_t^{(i)}, \mathbf{Y}_t^{(i)}\}_{t=1}^{T^{(i)}} \cup \{\mathbf{V}^{(i)}\} \right\}_{i=1}^{N}$. Each trajectory consists of sequential measurements over $T^{(i)}$ time points. At time $t$, $\mathbf{X}_t \in \mathcal{X}$ represents time-varying covariates, $\mathbf{A}_t \in [0,1]^d$ denotes continuous intervention, and $\mathbf{Y}_t \in \mathcal{Y}$ indicates measured outcome. Time-invariant characteristics are $\mathbf{V} \in \mathcal{V}$. We drop subject indicator $(i)$ for simplicity.

The problem is formulated as a finite-horizon MDP starting at time $t$. The state is $\mathbf{\Psi}_t = (\bar{\mathbf{H}}_t, \mathbf{Y}_{\text{target}})$, where $\bar{\mathbf{H}}_t = (\bar{\mathbf{X}}_t, \bar{\mathbf{A}}_{t-1}, \bar{\mathbf{Y}}_t, \mathbf{V})$ is the subject's full history. The bar notation represents sequences up to time $t$: $\bar{\mathbf{X}}_t = (\mathbf{X}_1, \cdots, \mathbf{X}_t)$, $\bar{\mathbf{Y}}_t = (\mathbf{Y}_1, \cdots, \mathbf{Y}_t)$, and $\bar{\mathbf{A}}_{t-1} = (\mathbf{A}_1, \cdots, \mathbf{A}_{t-1})$. $\mathbf{Y}_{\text{target}}$ is the desired outcome. Given target region $\mathcal{T} = \{\mathbf{y} : \|\mathbf{y} - \mathbf{Y}_{\text{target}}\| \leq \delta\}$, the reward function penalizes each step before reaching the target. For the $k$-th step into the future (at absolute time $t + k$), the reward is:

$$r_{t+k} = \begin{cases} 0, & \text{if } \mathbf{Y}_{t+k+1} \in \mathcal{T} \text{ and it is the first hit since time } t \\ -1, & \text{otherwise.} \end{cases} \tag{1}$$

A policy $\pi = (\pi_t, \ldots, \pi_{t+\tau_{\max}-1})$ is a sequence of decision rules for this future horizon. The value of a policy $\pi$ given the initial state $\psi_t = (\bar{\mathbf{h}}_t, \mathbf{y}_{\text{target}})$ is the expected sum of discounted future rewards:

$$V^\pi(\psi_t) = \mathbb{E}_\pi \left[ \sum_{k=0}^{\tau_{\max}-1} \gamma^k r_{t+k} \mid \mathbf{\Psi}_t = \psi_t \right], \tag{2}$$

where $\gamma \in (0, 1)$ is a discount factor. The objective is to find the optimal policy $\pi^*$ that maximizes this value function. This policy is composed of optimal actions at each future step $k \in [0, \tau_{\max} - 1]$, derived from the optimal action-value function $Q_{t+k}^*$:

$$\pi_{t+k}^*(\psi_{t+k}) = \operatorname{argmax}_{\mathbf{a} \in \mathcal{A}} Q_{t+k}^*(\psi_{t+k}, \mathbf{a}). \tag{3}$$

Learning this policy from observational data relies on standard causal inference assumptions, including consistency, sequential ignorability, and positivity; see Appendix B for details.

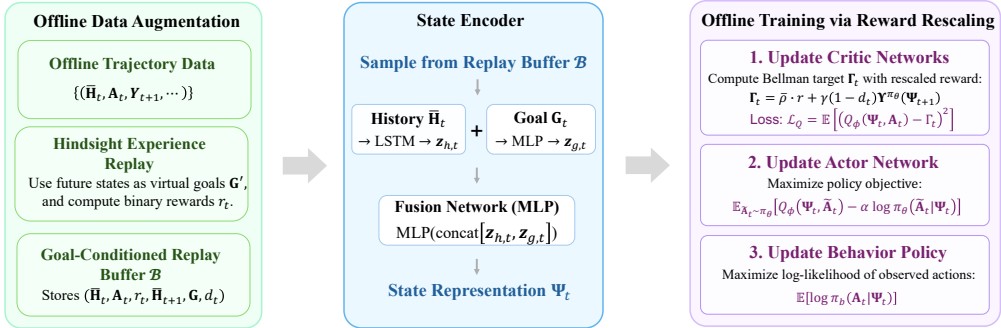

Figure 2: The overall architecture of the GIFT framework, encompassing three main stages: offline data preprocessing, state encoding, and the SAC training loop.

## 4 METHODOLOGY

### 4.1 STATE REPRESENTATION AND POLICY NETWORK ARCHITECTURE

GIFT learns dynamic intervention strategies by combining goal-conditioning with offline reinforcement learning (RL). The framework consists of three stages (Figure 2): (1) data augmentation using Hindsight Experience Replay; (2) state encoding fusing patient history and goals; and (3) SAC training with reward rescaling.

**State Representation.** We define the state at decision time $t$ as $\boldsymbol{\Psi}_t = (\mathbf{S}_t, \mathbf{G}_t)$, where $\mathbf{S}_t = \bar{\mathbf{H}}_t$ represents the patient's complete history and $\mathbf{G}_t = \mathbf{Y}_{\text{target}}$ represents the desired outcome. Concretely, $\bar{\mathbf{H}}_t$ is a variable-length sequence composed of past outcomes $\mathbf{Y}_{1:t}$, past interventions $\mathbf{A}_{1:t-1}$ (time-aligned), static covariates $\mathbf{V}$ repeated along time, and optionally vitals $\mathbf{X}_{1:t}$. We pad/pack the sequence and feed it to an LSTM to obtain a history embedding $\mathbf{z}_{h,t}$. The goal $\mathbf{Y}_{\text{target}}$ is mapped by a small MLP to a goal embedding $\mathbf{z}_{g,t}$, then concatenated with $\mathbf{z}_{h,t}$ and passed through a fusion MLP to produce the final state vector $\mathbf{z}_t$.

**Policy Architecture.** We employ SAC with actor network $\pi_\theta$ mapping composite state $\psi_t = (\mathbf{s}_t, \mathbf{g}t)$ to stochastic policy parameters, and critic network $Q\phi$ estimating soft action-value function $Q(\psi_t, \boldsymbol{a}_t)$. The actor takes $\mathbf{z}_t$ and outputs a continuous intervention $\mathbf{a}_t \in (0,1)^{d_a}$ via a sigmoid-squashed reparameterized Gaussian (the double-$Q$ critic consumes $[\mathbf{z}_t, \mathbf{a}_t]$). The intended network output is this normalized treatment vector. At prediction time, the policy is rolled out autoregressively: given the goal and current state, the network produces the next action $\mathbf{a}_{t+1}$.

### 4.2 HINDSIGHT EXPERIENCE REPLAY FOR OFFLINE DATA AUGMENTATION

To address the issue of sparse rewards, we adopt Hindsight Experience Replay (HER). The theoretical foundation of HER relies on a strict assumption of goal-independent dynamics.

**Assumption 4.1** (Decomposed Goal-Independent Dynamics)**.** *Physical state evolution depends only on the current state and action, independent of the goal, while the goal remains constant within transitions. The state transition probability can be decomposed as:*

$$P((\mathbf{S}_{t+1}, \mathbf{G}_{t+1}) = (\mathbf{s}', \mathbf{g}')|\mathbf{S}_t = \mathbf{s}, \mathbf{G}_t = \mathbf{g}, \mathbf{A}_t = \mathbf{a}) = P(\mathbf{S}_{t+1} = \mathbf{s}'|\mathbf{S}_t = \mathbf{s}, \mathbf{A}_t = \mathbf{a}) \cdot \mathbf{1}[\mathbf{g}' = \mathbf{g}]$$

Assumption 4.1 is plausible in dynamic intervention settings. In diabetes management, a patient's blood glucose response to insulin follows physiological laws, independent of the physician's long-term control objective. This independence allows us to generate new experiences by retrospectively relabeling goals.

For an observed transition $(\mathbf{s}_t, \mathbf{a}_t, r_t, \mathbf{s}_{t+1})$, we designate a hindsight goal $\mathbf{g}'$ sampled from future achieved outcomes of the same trajectory and compute $r'$ by recomputing the goal-conditioned reward, constructing $((\mathbf{s}_t, \mathbf{g}'), \mathbf{a}_t, r', (\mathbf{s}_{t+1}, \mathbf{g}'))$ (optionally sampling $k$ goals per transition) for data augmentation.

## 4.3 OFFLINE POLICY LEARNING VIA REWARD RESCALING

Training an RL agent on a fixed observational dataset is susceptible to distributional shift Levine et al. (2020). We employ a modified Bellman target to stabilize the learning process.

### 4.3.1 INTUITION OF REWARD RESCALING AS A HEURISTIC

Our core modification is to multiply the immediate reward $r(\boldsymbol{\psi}, \boldsymbol{a})$ by a clipped importance sampling ratio $\bar{\rho}(\boldsymbol{\psi}, \boldsymbol{a})$. Specifically, we first define the importance sampling ratio $\rho(\boldsymbol{\psi}, \boldsymbol{a}) = \frac{\pi_\theta(\boldsymbol{a}|\boldsymbol{\psi})}{\pi_b(\boldsymbol{a}|\boldsymbol{\psi})}$ and then clip it to a predefined interval $[\epsilon_1, \epsilon_2]$ to obtain $\bar{\rho}(\boldsymbol{\psi}, \boldsymbol{a}) = \text{clip}(\rho(\boldsymbol{\psi}, \boldsymbol{a}), \epsilon_1, \epsilon_2)$.

Intuitively, this operation optimizes a learning problem with a "rescaled reward" without altering environment dynamics. When the ratio exceeds 1, the actor policy prefers the action more than the behavior policy, amplifying its reward; the opposite occurs when the ratio is below 1. This skews the Bellman update target toward the actor policy's action distribution, counteracting learning signal bias from distribution mismatch.

This method's advantage is that weight adjustment within a limited range and application to single-step rewards avoids exploding variance from trajectory-level importance sampling. The evaluation operator remains a contraction, ensuring stable training. The trade-off introduces controllable approximation bias. Therefore, reward scaling provides a low-variance, bias-controlled policy-distribution re-weighting mechanism. We also consider Conservative Q-Learning (CQL (Kumar et al., 2020)) for distributional shift, which imposes critic-side conservatism via a log-sum-exp penalty on out-of-distribution actions. This contrasts with our clipped reward rescaling; empirical comparisons appear in Sec. 5.4.

### 4.3.2 FORMAL ANALYSIS AND THEORETICAL GUARANTEES

Our theoretical analysis aims to characterize the convergence and bias bounds of the algorithm under this heuristic modification, with its validity resting on the following assumptions.

**Assumption 4.2** (Bounded Function Class (Glivenko–Cantelli)). *The function class $\mathcal{Q}$ we use to approximate the Q-function is a bounded P-Glivenko–Cantelli class.*

As previously described, we use the policy-scaled reward function $\tilde{r}(\boldsymbol{\psi}, \boldsymbol{a}) = \bar{\rho}(\boldsymbol{\psi}, \boldsymbol{a})r(\boldsymbol{\psi}, \boldsymbol{a})$. Based on this, we proceed with the formal analysis.

**Definition 4.1** (Soft Policy Evaluation and Modified Bellman Operator). *For any Q-function $Q \in \mathcal{Q}$, we define the following operators:*

> ***Next-state Soft Value:*** $\quad \Upsilon^{\pi_\theta}(Q)(\boldsymbol{\psi}') = \mathbb{E}_{\mathbf{A}' \sim \pi_\theta(\cdot|\boldsymbol{\psi}')}[Q(\boldsymbol{\psi}', \mathbf{A}') - \alpha \log \pi_\theta(\mathbf{A}'|\boldsymbol{\psi}')],$
>
> ***Standard Operator:*** $\quad (\mathcal{T}^{\pi_\theta}Q)(\boldsymbol{\psi}, \boldsymbol{a}) = r(\boldsymbol{\psi}, \boldsymbol{a}) + \gamma\mathbb{E}_{\boldsymbol{\Psi}' \sim P(\cdot|\boldsymbol{\psi}, \boldsymbol{a})}[(1 - d(\boldsymbol{\Psi}'))\Upsilon^{\pi_\theta}(Q)(\boldsymbol{\Psi}')],$
>
> ***Modified Operator:*** $\quad (\mathcal{T}_{\tilde{\pi}}Q)(\boldsymbol{\psi}, \boldsymbol{a}) = \tilde{r}(\boldsymbol{\psi}, \boldsymbol{a}) + \gamma\mathbb{E}_{\boldsymbol{\Psi}' \sim P(\cdot|\boldsymbol{\psi}, \boldsymbol{a})}[(1 - d(\boldsymbol{\Psi}'))\Upsilon^{\pi_\theta}(Q)(\boldsymbol{\Psi}')],$

*where $d(\boldsymbol{\psi}') = 1$ if the episode terminates at state $\boldsymbol{\psi}'$, and 0 otherwise.*

To justify that our operator-based analysis is grounded in observable quantities, we first establish that the soft Bellman operator is causally identifiable from the observational data under standard assumptions.

**Proposition 4.1** (Causal Identifiability of the Soft Bellman Operator). *Assume Assumption 4.1 (goal-independent dynamics) and the standard assumptions detailed in Appendix B. Then, for any bounded $Q \in \mathcal{Q}$ and any state-action pair $(\boldsymbol{\psi}, \boldsymbol{a})$ at decision time $t$,*

$$(\mathcal{T}^{\pi_\theta} Q)(\boldsymbol{\psi}, \boldsymbol{a}) = r(\boldsymbol{\psi}, \boldsymbol{a}) + \gamma \, \mathbb{E}^{do(\pi_\theta)}_{\boldsymbol{\Psi}' \sim P(\cdot | \boldsymbol{\psi}, \boldsymbol{a})} \Big[ (1 - d(\boldsymbol{\Psi}')) \, \Upsilon^{\pi_\theta}(Q)(\boldsymbol{\Psi}') \Big], \tag{4}$$

$$= r(\boldsymbol{\psi}, \boldsymbol{a}) + \gamma \, \mathbb{E}_{\boldsymbol{\Psi}' \sim P_{obs}(\cdot | \bar{\mathbf{H}}_t, \mathbf{A}_t = \boldsymbol{a})} \Big[ (1 - d(\boldsymbol{\Psi}')) \, \mathbb{E}_{\mathbf{A}' \sim \pi_b(\cdot | \boldsymbol{\Psi}')} \big[ \rho(\boldsymbol{\Psi}', \mathbf{A}') \, \big( Q(\boldsymbol{\Psi}', \mathbf{A}')$$

$$- \alpha \log \pi_\theta(\mathbf{A}' | \boldsymbol{\Psi}') \big) \big] \Big]. \tag{5}$$

*Equation 4 is the soft Bellman evaluation operator under the interventional regime $do(\pi_\theta)$; Equation 5 shows that this interventional quantity is identifiable from the observational distribution $P_{obs}$ via one-step importance weighting with respect to the behavior policy $\pi_b$.*

With identifiability established, we now analyze the modified operator's contraction and the resulting convergence and bias bounds, which quantify the effect of reward rescaling via the clipped ratio.

**Theorem 4.2** (Contraction Property of the Modified Operator). *Let $\mathcal{T}_{\tilde{\pi}}$ be the modified Bellman operator. For any pair of bounded $Q$-functions $Q_1$ and $Q_2$, $\mathcal{T}_{\tilde{\pi}}$ satisfies:*

$$\|\mathcal{T}_{\tilde{\pi}} Q_1 - \mathcal{T}_{\tilde{\pi}} Q_2\|_\infty \leq \gamma \|Q_1 - Q_2\|_\infty$$

*Therefore, $\mathcal{T}_{\tilde{\pi}}$ is a contraction mapping with respect to the infinity norm $\| \cdot \|_\infty$ with a factor of $\gamma$.*

**Theorem 4.3** (Convergence and Asymptotic Performance Bound). *Under Assumptions 4.1–4.2, the Q-learning process converges to the unique fixed point $Q^{\tilde{\pi}}$ of the modified operator $\mathcal{T}_{\tilde{\pi}}$ (or to its projected fixed point in the case of function approximation). The gap between this fixed point and the true soft Q-function $Q^{\pi_\theta}$ of policy $\pi_\theta$ under the original reward $r$ is bounded as follows:*

$$\|Q^{\pi_\theta} - Q^{\tilde{\pi}}\|_\infty \leq \frac{1}{1 - \gamma} \|(\mathcal{T}^{\pi_\theta} - \mathcal{T}_{\tilde{\pi}}) Q^{\tilde{\pi}}\|_\infty$$

This bound clearly indicates that the gap between the solution $Q^{\tilde{\pi}}$ found by our algorithm and the true Q-value $Q^{\pi_\theta}$ of the actor policy is upper-bounded by the difference between the two operators evaluated at the fixed point $Q^{\tilde{\pi}}$. This difference, $\|(\mathcal{T}^{\pi_\theta} - \mathcal{T}_{\tilde{\pi}}) Q^{\tilde{\pi}}\|_\infty = \sup_{\boldsymbol{\psi}, \boldsymbol{a}} |r(\boldsymbol{\psi}, \boldsymbol{a})(1 - \bar{\rho}(\boldsymbol{\psi}, \boldsymbol{a}))|$, is precisely the systematic bias introduced by our heuristic reward rescaling.

## 5 EXPERIMENTS

### 5.1 EXPERIMENTAL SETUP

This section details the datasets, baseline models, and evaluation metrics used in our experiments to provide context for the subsequent analysis.

**Datasets.** We utilize two datasets for a comprehensive evaluation. The **Tumor** dataset is a classic simulated dataset based on a pharmacokinetic-pharmacodynamic framework (Geng et al., 2017). It allows us to control the influence of intervention history via a confounding parameter $\kappa$; larger $\kappa$ means more severe confounding, where past (including unobserved) factors jointly affect treatment and outcomes, increasing bias and hindering counterfactual prediction and policy learning. The other is a semi-synthetic experimental environment built upon the **MIMIC-III** database (Johnson et al., 2016). To construct a platform that mirrors real-world clinical complexity while maintaining controlled evaluation capabilities, we follow recent works (Hatt & Feuerriegel, 2024; Kuzmanovic et al., 2021; Melnychuk et al., 2022) and synthesize multi-dimensional outcomes and continuous interventions with complex temporal dependencies and confounding

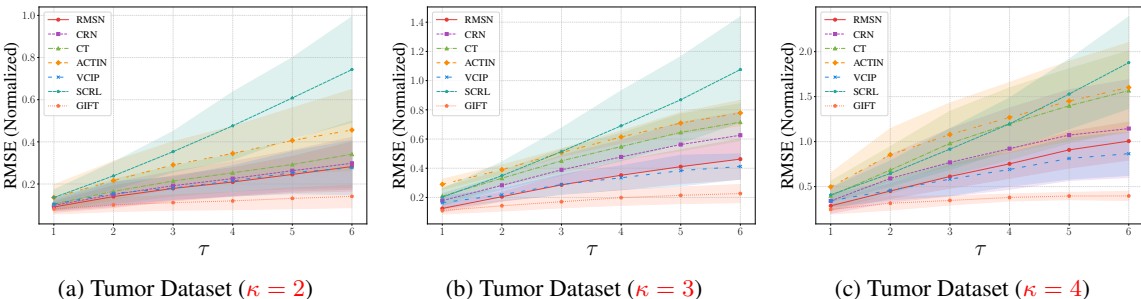

(a) Tumor Dataset ($\kappa = 2$)  (b) Tumor Dataset ($\kappa = 3$)  (c) Tumor Dataset ($\kappa = 4$)

Figure 3: Comparison of terminal-outcome RMSE at horizons $\tau \in 2, 3, 4$ for different models on the Tumor dataset under varying confounding levels $\kappa$. Each subplot shows mean RMSE with standard-deviation shading on the held-out test set; RMSE is computed between the terminal outcome after at most $\tau$ steps and the $\tau$-step target. This evaluation setting is used for all experiments unless otherwise noted.

relationships based on real ICU physiological data. This provides a robust testbed for evaluating model performance in more realistic scenarios. Intended clinical tasks: in Tumor, drive tumor burden (volume) to a desired target level/region; in the semi-synthetic MIMIC environment, steer a 2D clinical outcome vector $(y_1, y_2)$ toward a user-specified target $\mathbf{Y}_{target} \in \mathbb{R}^2$ within a tolerance, i.e., a goal-reaching control task over a planning horizon.

**Baselines and Decision Paradigms.** We compare against **RMSN** (Lim et al., 2018), **CRN** (Bica et al., 2020), **CT** (Melnychuk et al., 2022), **ACTIN** (Wang et al., 2024), **VCIP** (Wang et al., 2025), **SCRL** (Zheng et al., 2024). Since some baselines do not output policies, we use VCIP's episode-wise optimization and a step-wise greedy scheme at test time; both search action sequences to minimize target discrepancy. Results for episode-wise optimization are in the main text; step-wise details are in Appendix F, G. For SCRL, we adopt the official setup with minimal adaptations to tabular/time-series inputs (goal-conditioned actor + BC regularization, contrastive critic, large in-batch negatives, simple non-visual augmentations); tuning details appear in Appendix. F.

**Evaluation Metrics.** We adopt Root Mean Square Error (RMSE) as our primary evaluation metric, measuring discrepancy between the final trajectory from the model's generated intervention policy and the target trajectory. All reported values are mean and standard deviation over multiple independent runs. Concretely, for each test episode and a given horizon $\tau$, we roll out a policy for at most $\tau$ steps. Let $\mathbf{Y}_{term}$ be the terminal outcome at step $t+\tau$), and let $\mathbf{Y}_{target}$ be the goal defined by the protocol above. We compute $\text{RMSE} = \sqrt{\frac{1}{N} \sum_{i=1}^{N} \left\| \mathbf{Y}_{term}^{(i)} - \mathbf{Y}_{target}^{(i)} \right\|_2^2}$, where $N$ is the number of test episodes. This metric is defined on the terminal outcome only and it is computed on the held-out *test* set. Larger $\tau$ increases difficulty because errors compound over more transitions

## 5.2 COMPARATIVE EVALUATION OF INTERVENTION POLICIES

**Performance with Identical Intervention Strategies.** We first evaluate the performance of all models in a standard sequential decision-making setting, where the target states are within the support of the training data distribution. The results on the MIMIC-III synthetic dataset, presented in Table 2, show that GIFT outperforms all baseline models as measured by RMSE. Here "identical strategies" means training and testing targets are both induced by the behavior/original policy. We further corroborate this finding using the Tumor dataset under varying levels of confounding ($\kappa=2, 3, 4$), as shown in Figure 3. Larger $\kappa$ indicates more severe confounding; although all methods degrade as $\kappa$ increases, GIFT maintains the lowest RMSE.

Table 1: Results on tumor dataset ($\kappa = 4$) with distinct intervention strategies applied to training and test sets, reported as RMSE (mean $\pm$ std over five runs).

|  | $\tau = 1$ | $\tau = 2$ | $\tau = 3$ | $\tau = 4$ | $\tau = 5$ | $\tau = 6$ |
|---|---|---|---|---|---|---|
| RMSN | 0.32±0.04 | 0.53±0.08 | 0.69±0.09 | 0.85±0.14 | 1.01±0.19 | 1.15±0.21 |
| CRN | 0.41±0.15 | 0.71±0.28 | 0.92±0.39 | 1.10±0.46 | 1.26±0.50 | 1.40±0.51 |
| CT | 0.50±0.18 | 0.82±0.30 | 1.11±0.34 | 1.39±0.40 | 1.63±0.43 | 1.87±0.43 |
| ACTIN | 0.65±0.24 | 1.08±0.38 | 1.29±0.40 | 1.57±0.46 | 1.71±0.46 | 1.86±0.45 |
| VCIP | 0.46±0.09 | 0.66±0.18 | 0.76±0.13 | 0.89±0.13 | 0.99±0.20 | 1.10±0.20 |
| SCRL | 0.53±0.09 | 0.91±0.24 | 1.23±0.25 | 1.56±0.19 | 1.98±0.26 | 2.42±0.38 |
| **GIFT** | **0.24±0.04** | **0.32±0.04** | **0.38±0.05** | **0.43±0.08** | **0.45±0.07** | **0.46±0.08** |

Table 2: Results on MIMIC synthetic dataset with the same intervention strategies applied to training and test sets, reported as RMSE (mean $\pm$ std over five runs).

|  | $\tau = 1$ | $\tau = 2$ | $\tau = 3$ | $\tau = 4$ | $\tau = 5$ | $\tau = 6$ |
|---|---|---|---|---|---|---|
| RMSN | 0.25±0.07 | 0.39±0.13 | 0.50±0.17 | 0.60±0.20 | 0.70±0.21 | 0.83±0.28 |
| CRN | 0.31±0.04 | 0.46±0.11 | 0.60±0.15 | 0.71±0.17 | 0.83±0.18 | 0.96±0.24 |
| CT | 0.62±0.17 | 1.06±0.32 | 1.42±0.47 | 1.72±0.60 | 1.98±0.73 | 2.20±0.83 |
| ACTIN | 0.28±0.19 | 0.57±0.34 | 0.79±0.50 | 0.98±0.64 | 1.11±0.74 | 1.23±0.81 |
| VCIP | 0.41±0.19 | 0.51±0.22 | 0.57±0.21 | 0.60±0.20 | 0.62±0.18 | 0.67±0.22 |
| SCRL | 0.37±0.18 | 0.47±0.21 | 0.56±0.26 | 0.59±0.28 | 0.63±0.30 | 0.65±0.30 |
| **GIFT** | **0.24±0.10** | **0.31±0.13** | **0.33±0.14** | **0.37±0.16** | **0.38±0.17** | **0.39±0.18** |

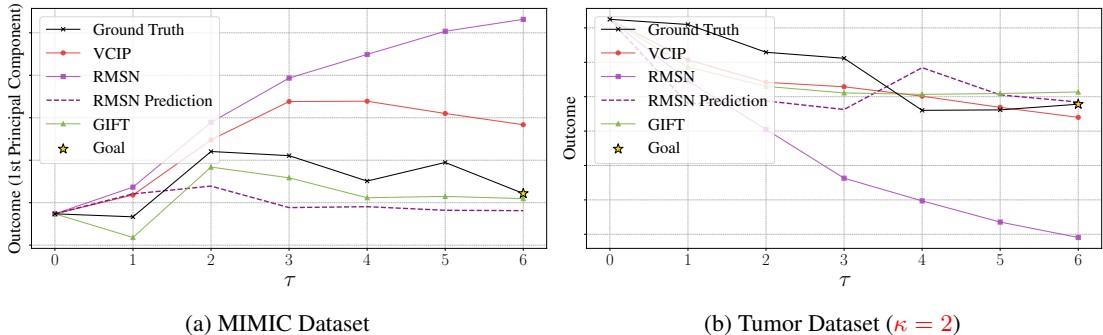

(a) MIMIC Dataset         (b) Tumor Dataset ($\kappa = 2$)

Figure 4: Case study of intervention policies from GIFT and baselines. The figure shows realized outcome trajectories for one patient on (a) MIMIC and (b) Tumor ($\kappa = 2$). 'Goal' is the $\tau$-step target and is marked by a gold star at $\tau = 6$. 'RMSN Prediction' is the purple dashed line; other model trajectories are solid.

**Generalization to Unseen Intervention Strategies.** Next, we assess the ability to reach targets generated by intervention rules unseen during training. "Unseen strategies" are created by overriding, with probability $\eta$ at each step, the standard assignment using actions sampled from a static $\mathrm{Beta}(\alpha, \beta)$ distribution (Appendix C). The results on the Tumor dataset (at $\kappa$=4), detailed in Table 1, show that GIFT achieves the lowest error, demonstrating strong generalization.

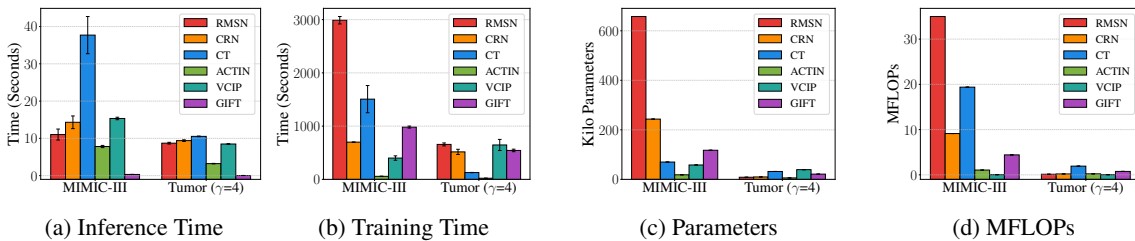

Figure 5: Efficiency comparison of the GIFT model with alternative models on the MIMIC-III and Tumor ($\kappa = 4$) datasets. The comparison is conducted across four metrics: (a) number of parameters, (b) MFLOPs, (c) training time, and (d) inference time. The results are presented as the average over multiple runs.

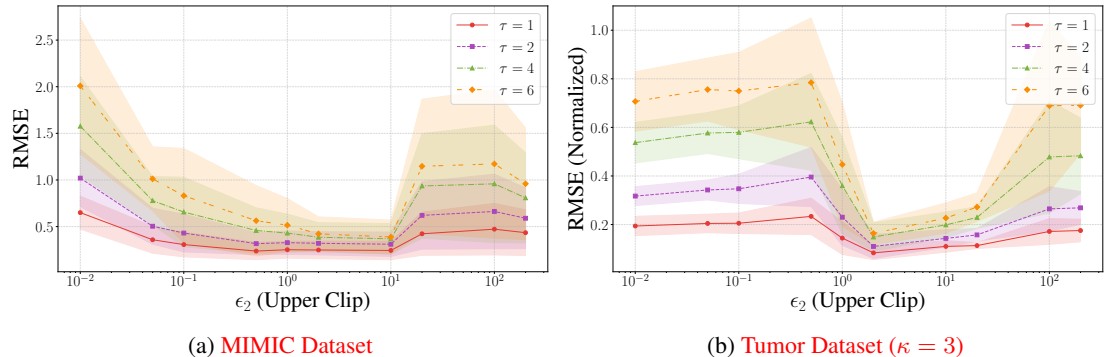

Figure 6: Effect of importance-weight clipping on performance. We fix $\epsilon_1 = 0.01$ and vary the upper clip $\epsilon_2 \in \{0.01, 0.05, 0.1, 0.5, 1.0, 2.0, 10.0, 20, 100, 200\}$. Curves show RMSE versus $\epsilon_2$ for (a) MIMIC and (b) Tumor datasets across $\tau \in \{1, 2, 4, 6\}$. For Tumor, RMSE is normalized as indicated in the figure.

## 5.3 QUALITATIVE AND EFFICIENCY ANALYSIS

**Qualitative Analysis: A Case Study.** To visually demonstrate GIFT's effectiveness, we conduct a case study. Figure 4 compares predicted outcome trajectories for a single patient generated by GIFT and baseline models. The figure plots 'Ground Truth' and 'Goal' trajectories, illustrating how well each model steers the patient toward the desired target. Counterfactual prediction models suffer performance degradation from accumulated prediction errors; their intervention optimization relies on imperfect forecasts that fail to reflect true outcome trends. In contrast, GIFT's learned policy proves more robust, guiding the patient's trajectory more effectively toward the goal.

**Computational Efficiency.** Beyond efficacy, computational efficiency is crucial for practical deployment. We compare GIFT's computational costs against baselines across four dimensions: parameters, MFLOPs, training time, and inference time (Figure 5). Results indicate GIFT is competitive across multiple efficiency metrics. Most critically, because GIFT avoids iterative optimization during inference, its inference efficiency significantly exceeds baseline methods that rely on time-consuming search procedures. This makes GIFT well-suited for real-time online tasks demanding rapid decision-making.

Table 3: Ablation study results for different model configurations. Lower RMSE values are better, with the best performance in each column highlighted in **bold**.

| | MIMIC-III | | Tumor ($\kappa = 2$) | | Tumor ($\kappa = 4$) | |
|---|---|---|---|---|---|---|
| | $\tau = 3$ | $\tau = 6$ | $\tau = 3$ | $\tau = 6$ | $\tau = 3$ | $\tau = 6$ |
| Full Model | **0.33±0.14** | **0.39±0.18** | **0.11±0.04** | **0.14±0.05** | **0.35±0.05** | **0.40±0.05** |
| w/o RR | 0.66±0.39 | 0.83±0.58 | 0.48±0.17 | 0.62±0.27 | 0.49±0.13 | 0.42±0.15 |
| w/o Her | 0.66±0.49 | 0.95±0.80 | 0.23±0.07 | 0.40±0.15 | 0.80±0.24 | 1.53±0.53 |
| with CQL | 0.45±0.19 | 0.49±0.20 | 0.24±0.07 | 0.33±0.07 | 0.69±0.15 | 1.04±0.27 |

## 5.4 Sensitivity and Ablation Analysis

This section aims to dissect the internal mechanisms contributing to GIFT's success and to explore its performance under varying data conditions, thereby validating the rationale behind our model's design.

**Analysis of Importance-Weight Clipping.**

We study how the clipping interval $[\epsilon_1, \epsilon_2]$ in reward rescaling affects performance. We fix $\epsilon_1 = 0.01$ and sweep $\epsilon_2 \in \{0.01, 0.05, 0.1, 0.5, 1.0, 2.0, 10.0, 20, 100, 200\}$. Figure 6 shows RMSE vs. $\epsilon_2$ on MIMIC and Tumor ($\kappa = 3$) across $\tau \in \{1, 2, 4, 6\}$. Tight clips ($\epsilon_2 \leq 0.1$) induce truncation bias (higher RMSE); very loose clips ($\epsilon_2 \geq 20$) increase variance and instability. A broad sweet spot lies in $[1, 10]$, with $\epsilon_2 \approx 2$ near-optimal for most $\tau$. Larger $\tau$ raises difficulty and sensitivity.

**Ablation Study of Key Model Components.** To validate our proposed components, we conducted an ablation study (Table 3). This compares the full model against variants without Reward Rescaling ("w/o RR") and without Hindsight Experience Replay ("w/o HER"). Results demonstrate that removing either component significantly degrades performance, confirming their necessity. Comparing performance standard deviations reveals the variance-reducing benefit of our mechanism. On MIMIC-III with $\tau = 3$, standard deviation increases from 0.14 to 0.39, providing strong evidence that Reward Rescaling reduces training variance, enabling stable intervention policies. Additionally, we include a conservative offline RL control substituting our rescaling with CQL ("with CQL"). While CQL mitigates extrapolation error relative to "w/o RR", it remains inferior across datasets and horizons, indicating critic-only conservatism is less aligned with target attainment than our clipped reward re-weighting. Our full model exhibits smaller variance, especially under stronger confounding ($\kappa = 4$).

## 6 Conclusion

This work introduces GIFT, a novel offline framework for deriving sequential intervention policies from observational data. By formulating counterfactual target achievement as a goal-conditioned MDP, GIFT overcomes limitations of traditional offline planning. It addresses sparse rewards and distributional shift by integrating HER and a variance-controlled reward rescaling mechanism. Supported by convergence guarantees, extensive experiments show GIFT markedly surpasses existing methods in generating effective, generalizable, and computationally efficient policies. Its superior performance, particularly low inference cost, underscores broad suitability for real-time, adaptive decision-making in critical applications like personalized medicine and other high-stakes settings.

ETHICS STATEMENT

We acknowledge the ICLR Code of Ethics. This study uses synthetic and semi-synthetic data derived from de-identified MIMIC-III, reducing privacy risks and avoiding identifiable personal information. Learning policies from observational data may reflect dataset biases, unobserved confounding, and distribution shift, affecting fairness and reliability; we analyze sensitivity to confounding and goal sparsity and discuss limitations to avoid overstating clinical readiness. We note potential dual-use risks (e.g., optimizing harmful objectives) and emphasize that real-world use should include appropriate safeguards and expert oversight. We declare no conflicts of interest and credit prior work to support transparency and research integrity.

REPRODUCIBILITY STATEMENT

We are committed to ensuring the reproducibility of our research. To this end, we have made our source code and a detailed reproduction guide available in the **supplementary materials**. For a comprehensive description of the datasets used in our experiments, including details on data processing and splits, please refer to Appendix C. Furthermore, we provide a thorough discussion of our hyperparameter tuning process and the final selected values in Appendix D. We believe that these resources will enable the research community to fully reproduce our results and build upon our work.

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

## A    EXTENDED RELATED WORK

The problem of identifying an optimal sequence of interventions from offline data to achieve a target outcome intersects causal inference, sequential decision-making, and reinforcement learning. We group prior work into two complementary paradigms: (i) learn-then-plan with counterfactual world models, and (ii) goal-conditioned reinforcement learning. We also connect to the literature on Dynamic Treatment Regimes (DTRs).

### A.1    OFFLINE PLANNING FOR TARGET ACHIEVEMENT

A common workflow for target achievement is a two-stage "learn-then-plan" approach. First, a counterfactual world model is trained on observational data to predict outcomes under hypothetical interventions. Second, at inference time, an optimization procedure searches over prospective intervention sequences on top of this model to reach a user-specified target. The modeling component has roots in classical methods such as the g-formula and marginal structural models (MSMs) (Robins, 1986; Robins et al., 2000; Fitzmaurice et al., 2008). While these frameworks provide principled identification strategies, practical deployment in high-dimensional, nonlinear, and long-horizon settings can be challenging, which has motivated deep learning approaches.

Deep counterfactual sequence models focus on representation learning and deconfounding for longitudinal data. Representative examples include RMSN (Lim et al., 2018), which leverages inverse-probability weighting within a recurrent architecture; CRN (Bica et al., 2020), which learns adversarially balanced representations; and the Causal Transformer (CT) (Melnychuk et al., 2022), which introduces Transformer-based sequence modeling (Vaswani et al., 2017) to better capture long-range dependencies (Hochreiter et al., 2001). Related work also explores g-computation with deep networks (Li et al., 2021) and Bayesian/nonparametric approaches for longitudinal causal inference (Raveendran et al., 2020; Soleimani et al., 2017). In practice, these counterfactual models can be paired with a separate planning step that optimizes a target-specific objective on the learned world model. Distinct from these modeling papers, recent planning-oriented formulations explicitly cast target attainment as an optimization problem over the learned dynamics, including dual-module architectures for temporal counterfactual estimation (Wang et al., 2024) and maximum-likelihood-style formulations (Wang et al., 2025).

When a separate optimization is performed at inference time on top of a learned model, the resulting plans are often open-loop and can require iterative search, which may be sensitive to model misspecification and environmental stochasticity. Our work departs from purely open-loop planning by learning a reactive, closed-loop policy that conditions on both the current state and the target.

### A.2    GOAL-CONDITIONED REINFORCEMENT LEARNING

Goal-conditioned reinforcement learning (RL) directly learns a policy $\pi(a \mid s, g)$ that maps a state $s$ and goal $g$ to an action, thereby embedding planning into the policy itself and avoiding test-time trajectory optimization (Levine et al., 2020). A central challenge is sparse or delayed rewards. Hindsight Experience Replay (HER) (Andrychowicz et al., 2017) addresses this by relabeling unsuccessful trajectories with achieved goals, substantially improving sample efficiency.

Hierarchical Reinforcement Learning (HRL) is often employed for complex, goal-oriented tasks with sparse rewards. A Causality-Driven HRL (CDHRL) framework that discovers effective subgoal hierarchy structures has been introduced (Hu et al., 2022). This approach was evaluated in complex game environments such as 2d-Minecraft (Sohn et al., 2018) and Eden (Chen et al., 2021). For zero-shot transfer in object-oriented planning, Schema Networks have been developed as generative causal models that allow an agent to reason backward through causes to achieve goals (Kansky et al., 2017).

Many applications of these methods are found in robotics, where tasks are inherently goal-conditioned. For instance, causal reasoning has been used to learn and transfer robot manipulation policies (Lee et al., 2021). Tasks where a robot must reach a target location or move at a target speed have been considered (Feng et al., 2022). Similarly, an invariant policy optimization algorithm has been developed for a challenging task where a robot navigates to a key, opens a door, and proceeds to a final goal (Sonar et al., 2021). To facilitate research in this area, the *CausalWorld* benchmark was developed to test causal structure learning and transfer learning for robotic manipulation tasks where objects must be moved to specified goal locations (Ahmed et al., 2020). To our knowledge, our work is the first to formulate the counterfactual target achievement problem as an offline, goal-conditioned task.

Beyond classical formulations, recent GCRL advances emphasize learning from diverse offline or reward-free data for long-horizon goal reaching. C-Learning reframes goal-conditioned control as density estimation via a future versus non-future classifier, enabling off-policy prediction of a new policy's future state distribution and optimizing the probability of hitting a goal set (Eysenbach et al., 2020). Distributional Distance Classifiers bridge the tension between maximizing success probability and minimizing expected steps by estimating the probability of reaching the goal at different future timesteps and propose a practical Distributional NCE estimator (Akella et al., 2023). HIQL introduces a hierarchical offline GCRL method that treats latent states as actions, decomposes distant goal reaching into subgoal selection and low-level control, and leverages action-free data (Park et al., 2023). In robotics, Stabilizing Contrastive RL shows that contrastive self-supervised objectives together with careful architectural and hyperparameter choices can stabilize offline goal reaching and enable real-world image-based manipulation with a single goal image provided after training (Zheng et al., 2024).

These methods target general goal reaching and typically do not address confounding in observational data. Our setting, namely SCTA, requires learning a closed-loop policy from observational and potentially confounded trajectories to steer outcomes into a target region $T$. This demands both causal identifiability under standard assumptions and stability under offline distribution shift between the behavior policy $\pi_b$ and the learned policy $\pi_\theta$. Our framework (GIFT) combines goal conditioning with HER and clipped-importance reward rescaling to obtain a modified soft Bellman operator that remains a contraction and admits explicit bias bounds. In contrast to many GCRL systems that rely on test-time optimization or assume reward-free yet unconfounded data, GIFT provides a theoretically grounded and offline-stable learning procedure tailored to SCTA, where counterfactual identifiability and robustness to the shift from $\pi_b$ to $\pi_\theta$ are primary considerations.

### A.3 DYNAMIC TREATMENT REGIMES

Dynamic Treatment Regimes (DTRs) formalize sequential decision-making via a sequence of decision rules mapping evolving patient histories to interventions (Murphy, 2003). Reinforcement learning has been widely adopted to optimize DTRs from observational data (Zhang & Bareinboim, 2019; Luckett et al., 2020), including settings with continuous states and clinical constraints (Raghu et al., 2017). The standard DTR objective is to learn a single policy that maximizes population-level expected outcomes.

In contrast, our problem is explicitly goal-conditioned: rather than maximizing an undirected cumulative reward, we learn policies that steer individual trajectories toward a predefined target state. This goal-centric perspective complements the DTR paradigm by emphasizing targeted control for individual endpoints, while retaining the benefits of offline learning and robustness to confounding.

# B PROOFS

**Assumption B.1** (Decomposed Goal-Independent Dynamics). *Let $\mathbf{S}_t$ be the random variable for the physical state at time $t$, and $\mathbf{G}_t$ be the goal. The state transition probability can be decomposed as:*

$$P((\mathbf{S}_{t+1}, \mathbf{G}_{t+1}) = (\mathbf{s}', \mathbf{g}')|\mathbf{S}_t = \mathbf{s}, \mathbf{G}_t = \mathbf{g}, \mathbf{A}_t = \mathbf{a}) = P(\mathbf{S}_{t+1} = \mathbf{s}'|\mathbf{S}_t = \mathbf{s}, \mathbf{A}_t = \mathbf{a}) \cdot \mathbf{1}[\mathbf{g}' = \mathbf{g}]$$

*This implies that the evolution of the physical state, $P(\mathbf{S}_{t+1}|\mathbf{S}_t, \mathbf{A}_t)$, depends only on the current physical state and action, and is independent of the goal $\mathbf{G}_t$. Concurrently, the goal remains constant within a single-step transition.*

**Assumption B.2** (Consistency). *If the actual treatment decision is $\mathbf{A}_t = \mathbf{a}$, then the potential outcome is consistent with the observed outcome, i.e.,*

$$\mathbf{Y}_{t+1}[\mathbf{a}] = \mathbf{Y}_{t+1}, \quad r_t[\mathbf{a}] = r_t.$$

**Assumption B.3** (Sequential Overlap). *The behavior policy $\pi_b$ is sufficiently stochastic within its support. Specifically, for any state $\psi$ and action $\mathbf{a}$, if an action is possible under the behavior policy (i.e., $\pi_b(\mathbf{a}|\psi) > 0$), then its probability density is bounded below by a positive constant. That is, there exists a constant $c > 0$ such that $\pi_b(\mathbf{a}|\psi) \geq c$.*

**Assumption B.4** (Sequential Ignorability). *For any action $\mathbf{a}$, the current treatment decision $\mathbf{A}_t$ is independent of the future potential outcomes $\mathbf{Y}_{t+1}[\mathbf{a}]$, conditional on the past history $\mathbf{\Psi}_t$, i.e.:*

$$\mathbf{A}_t \perp\!\!\!\perp \mathbf{Y}_{t+1}[\mathbf{a}] \mid \mathbf{\Psi}_t.$$

**Assumption B.5** (Bounded Function Class (Glivenko–Cantelli)). *The function class $\mathcal{Q}$ we use to approximate the Q-function is a bounded P-Glivenko–Cantelli class.*

**Definition B.1** (Soft Policy Evaluation and Modified Bellman Operator). *For any Q-function $Q \in \mathcal{Q}$, we define the following operators:*

> ***Next-state Soft Value:*** $\quad \Upsilon^{\pi_\theta}(Q)(\psi') = \mathbb{E}_{\mathbf{A}' \sim \pi_\theta(\cdot|\psi')}[Q(\psi', \mathbf{A}') - \alpha \log \pi_\theta(\mathbf{A}'|\psi')]$

> ***Standard Operator:*** $\quad (\mathcal{T}^{\pi_\theta} Q)(\psi, \mathbf{a}) = r(\psi, \mathbf{a}) + \gamma \mathbb{E}_{\mathbf{\Psi}' \sim P(\cdot|\psi, \mathbf{a})}[(1 - d(\mathbf{\Psi}'))\Upsilon^{\pi_\theta}(Q)(\mathbf{\Psi}')]$

> ***Modified Operator:*** $\quad (\mathcal{T}_{\tilde{\pi}} Q)(\psi, \mathbf{a}) = \tilde{r}(\psi, \mathbf{a}) + \gamma \mathbb{E}_{\mathbf{\Psi}' \sim P(\cdot|\psi, \mathbf{a})}[(1 - d(\mathbf{\Psi}'))\Upsilon^{\pi_\theta}(Q)(\mathbf{\Psi}')]$

*where $d(\psi')$ is an indicator function. The expectation $\mathbb{E}_{\mathbf{\Psi}' \sim P(\cdot|\psi, \mathbf{a})}$ should be understood as an expectation over the next physical state $\mathbf{S}'$, as the goal remains unchanged during the transition.*

**Proposition B.1** (Causal Identifiability of the Soft Bellman Operator). *Assume Assumption B.1 (goal-independent dynamics) and the standard assumptions detailed in Appendix B. Then, for any bounded $Q \in \mathcal{Q}$ and any state-action pair $(\boldsymbol{\psi}, \boldsymbol{a})$ at decision time $t$,*

$$(\mathcal{T}^{\pi_\theta} Q)(\boldsymbol{\psi}, \boldsymbol{a}) = r(\boldsymbol{\psi}, \boldsymbol{a}) + \gamma \, \mathbb{E}^{do(\pi_\theta)}_{\mathbf{\Psi}' \sim P(\cdot|\boldsymbol{\psi}, \boldsymbol{a})}\Big[(1 - d(\mathbf{\Psi}')) \, \Upsilon^{\pi_\theta}(Q)(\mathbf{\Psi}')\Big], \tag{6}$$

$$= r(\boldsymbol{\psi}, \boldsymbol{a}) + \gamma \, \mathbb{E}_{\mathbf{\Psi}' \sim P_{obs}(\cdot|\bar{\mathbf{H}}_t, \mathbf{A}_t = \boldsymbol{a})}\Big[(1 - d(\mathbf{\Psi}')) \, \mathbb{E}_{\mathbf{A}' \sim \pi_b(\cdot|\mathbf{\Psi}')}\big[\rho(\mathbf{\Psi}', \mathbf{A}') \left(Q(\mathbf{\Psi}', \mathbf{A}')\right.$$

$$\left. - \alpha \log \pi_\theta(\mathbf{A}' \mid \mathbf{\Psi}'))\big]\Big]. \tag{7}$$

*Equation 6 is the soft Bellman evaluation operator under the interventional regime $do(\pi_\theta)$; Equation 7 shows that this interventional quantity is identifiable from the observational distribution $P_{obs}$ via one-step importance weighting with respect to the behavior policy $\pi_b$.*

*Proof.* We provide a do-calculus based identification in three steps.

**Step 1:** Fix the composite state $\mathbf{\Psi}_t = (\bar{\mathbf{H}}_t, \mathbf{Y}_{\text{target}})$ and action $\boldsymbol{a}_t = \boldsymbol{a}$ at time $t$. Under the intervention $do(\pi_\theta)$, actions are drawn from the target policy while the environment dynamics remain unchanged. By Definition 4.1,

$$(\mathcal{T}^{\pi_\theta} Q)(\boldsymbol{\psi}_t, \boldsymbol{a}) := r(\boldsymbol{\psi}_t, \boldsymbol{a}) + \gamma \, \mathbb{E}_{\mathbf{\Psi}_{t+1} \sim P(\cdot | \boldsymbol{\psi}_t, \boldsymbol{a})}^{do(\pi_\theta)} \Big[ (1 - d(\mathbf{\Psi}_{t+1})) \, \Upsilon^{\pi_\theta}(Q)(\mathbf{\Psi}_{t+1}) \Big], \tag{8}$$

where $\Upsilon^{\pi_\theta}(Q)(\boldsymbol{\psi}') = \mathbb{E}_{\mathbf{A}' \sim \pi_\theta(\cdot | \boldsymbol{\psi}')} \big[ Q(\boldsymbol{\psi}', \mathbf{A}') - \alpha \log \pi_\theta(\mathbf{A}' \mid \boldsymbol{\psi}') \big]$.

**Step 2:** Let the potential outcomes be $\mathbf{Y}_{t+1}(\boldsymbol{a})$ and the next physical state $\mathbf{S}_{t+1}(\boldsymbol{a})$. Consistency implies that whenever $\mathbf{A}_t = \boldsymbol{a}$,

$$\mathbf{Y}_{t+1} = \mathbf{Y}_{t+1}(\boldsymbol{a}), \qquad \mathbf{S}_{t+1} = \mathbf{S}_{t+1}(\boldsymbol{a}).$$

Sequential Ignorability (no unmeasured confounding) given $\bar{\mathbf{H}}_t$ implies

$$\{\mathbf{Y}_{t+1}(\boldsymbol{a}), \mathbf{S}_{t+1}(\boldsymbol{a})\} \perp\!\!\!\perp \mathbf{A}_t \mid \bar{\mathbf{H}}_t, \quad \forall \boldsymbol{a}.$$

By the back-door criterion (do-calculus Rule 2) or the g-formula,

$$P\big(\mathbf{S}_{t+1} \in \cdot \mid do(\mathbf{A}_t = \boldsymbol{a}), \bar{\mathbf{H}}_t\big) = P\big(\mathbf{S}_{t+1} \in \cdot \mid \bar{\mathbf{H}}_t, \mathbf{A}_t = \boldsymbol{a}\big), \tag{9}$$

$$P\big(\mathbf{Y}_{t+1} \in \cdot \mid do(\mathbf{A}_t = \boldsymbol{a}), \bar{\mathbf{H}}_t\big) = P\big(\mathbf{Y}_{t+1} \in \cdot \mid \bar{\mathbf{H}}_t, \mathbf{A}_t = \boldsymbol{a}\big). \tag{10}$$

Assumption 4.1 (goal-independent dynamics) guarantees that within one step the goal component in $\mathbf{\Psi}_{t+1} = (\bar{\mathbf{H}}_{t+1}, \mathbf{Y}_{\text{target}})$ remains constant and the next physical state depends only on $(\bar{\mathbf{H}}_t, \mathbf{A}_t)$. Therefore, the outer expectation in equation 8 can be written in terms of the observational conditional distribution $P_{obs}(\cdot \mid \bar{\mathbf{H}}_t, \mathbf{A}_t = \boldsymbol{a})$, which yields equation 6.

**Step 3:** Conditioned on $\mathbf{\Psi}'$, the inner expectation in $\Upsilon^{\pi_\theta}(Q)(\mathbf{\Psi}')$ can be expressed under the behavior policy $\pi_b$ using the importance ratio $\rho(\mathbf{\Psi}', \mathbf{A}') = \frac{\pi_\theta(\mathbf{A}' | \mathbf{\Psi}')}{\pi_b(\mathbf{A}' | \mathbf{\Psi}')}$, provided Positivity holds. For any integrable function $g$,

$$\mathbb{E}_{\mathbf{A}' \sim \pi_\theta(\cdot | \mathbf{\Psi}')}[g(\mathbf{A}')] = \mathbb{E}_{\mathbf{A}' \sim \pi_b(\cdot | \mathbf{\Psi}')} \Big[ \rho(\mathbf{\Psi}', \mathbf{A}') \, g(\mathbf{A}') \Big].$$

Taking $g(\mathbf{A}') = Q(\mathbf{\Psi}', \mathbf{A}') - \alpha \log \pi_\theta(\mathbf{A}' \mid \mathbf{\Psi}')$ gives

$$\Upsilon^{\pi_\theta}(Q)(\mathbf{\Psi}') = \mathbb{E}_{\mathbf{A}' \sim \pi_b(\cdot | \mathbf{\Psi}')} \Big[ \rho(\mathbf{\Psi}', \mathbf{A}') \big( Q(\mathbf{\Psi}', \mathbf{A}') - \alpha \log \pi_\theta(\mathbf{A}' \mid \mathbf{\Psi}') \big) \Big].$$

Substituting this back into equation 6 and using equation 9–equation 10 yields equation 7. Hence, under the stated causal assumptions, $\mathcal{T}^{\pi_\theta}$ is identifiable from the observational distribution via one-step importance weighting.

**Remark on clipping.** Replacing $\rho$ by its clipped version $\bar{\rho} = \text{clip}(\rho, \epsilon_1, \epsilon_2)$ produces a controlled-bias approximation of equation 7. The bias introduced by clipping is precisely quantified by the operator gap term appearing in the subsequent bound, $\sup_{\boldsymbol{\psi}, \boldsymbol{a}} |r(\boldsymbol{\psi}, \boldsymbol{a})(1 - \bar{\rho}(\boldsymbol{\psi}, \boldsymbol{a}))|/(1 - \gamma)$, thus enabling a bias–variance trade-off while preserving the identifiable structure of the operator. $\square$

**Theorem B.2** (Contraction Property of the Modified Operator). *Let $\mathcal{T}_{\tilde{\pi}}$ be the modified Bellman operator defined in Definition B.1. For any pair of bounded Q-functions $Q_1$ and $Q_2$, $\mathcal{T}_{\tilde{\pi}}$ satisfies:*

$$\|\mathcal{T}_{\tilde{\pi}} Q_1 - \mathcal{T}_{\tilde{\pi}} Q_2\|_\infty \leq \gamma \|Q_1 - Q_2\|_\infty$$

*Therefore, $\mathcal{T}_{\tilde{\pi}}$ is a contraction mapping with respect to the infinity norm $\|\cdot\|_\infty$ with a factor of $\gamma$.*

*Proof.* For any two Q-functions $Q_1, Q_2 \in \mathcal{Q}$ and any state-action pair $(\psi, \mathbf{a})$, we examine the difference after applying the operator. According to the operator's definition, the reward term $\tilde{r}$ is canceled out in the subtraction:

$$|(\mathcal{T}_{\tilde{\pi}}Q_1)(\psi, \mathbf{a}) - (\mathcal{T}_{\tilde{\pi}}Q_2)(\psi, \mathbf{a})| = \left|\gamma\mathbb{E}_{\mathbf{\Psi}' \sim P(\cdot|\psi, \mathbf{a})}[(1 - d(\mathbf{\Psi}'))(\Upsilon^{\pi_\theta}(Q_1)(\mathbf{\Psi}') - \Upsilon^{\pi_\theta}(Q_2)(\mathbf{\Psi}'))]\right|$$

$$\leq \gamma\mathbb{E}_{\mathbf{\Psi}' \sim P(\cdot|\psi, \mathbf{a})}\left[|(1 - d(\mathbf{\Psi}'))(\Upsilon^{\pi_\theta}(Q_1)(\mathbf{\Psi}') - \Upsilon^{\pi_\theta}(Q_2)(\mathbf{\Psi}'))|\right]$$

$$\leq \gamma\mathbb{E}_{\mathbf{\Psi}' \sim P(\cdot|\psi, \mathbf{a})}\left[|\Upsilon^{\pi_\theta}(Q_1)(\mathbf{\Psi}') - \Upsilon^{\pi_\theta}(Q_2)(\mathbf{\Psi}')|\right]$$

Next, we analyze the difference of the inner term. By the definition of the next-state soft value:

$$|\Upsilon^{\pi_\theta}(Q_1)(\psi') - \Upsilon^{\pi_\theta}(Q_2)(\psi')| = \left|\mathbb{E}_{\mathbf{A}' \sim \pi_\theta(\cdot|\psi')}[Q_1(\psi', \mathbf{A}') - Q_2(\psi', \mathbf{A}')]\right|$$

$$\leq \mathbb{E}_{\mathbf{A}' \sim \pi_\theta(\cdot|\psi')}[|Q_1(\psi', \mathbf{A}') - Q_2(\psi', \mathbf{A}')|]$$

$$\leq \mathbb{E}_{\mathbf{A}' \sim \pi_\theta(\cdot|\psi')}\left[\sup_{\psi^*, \mathbf{a}^*} |Q_1(\psi^*, \mathbf{a}^*) - Q_2(\psi^*, \mathbf{a}^*)|\right]$$

$$= \|Q_1 - Q_2\|_\infty$$

Substituting this upper bound back into the first inequality, we obtain:

$$|(\mathcal{T}_{\tilde{\pi}}Q_1)(\psi, \mathbf{a}) - (\mathcal{T}_{\tilde{\pi}}Q_2)(\psi, \mathbf{a})| \leq \gamma\mathbb{E}_{\mathbf{\Psi}' \sim P(\cdot|\psi, \mathbf{a})}[\|Q_1 - Q_2\|_\infty] = \gamma\|Q_1 - Q_2\|_\infty$$

Since this inequality holds for all $(\psi, \mathbf{a})$, we can take the supremum over all pairs, which proves that $\|\mathcal{T}_{\tilde{\pi}}Q_1 - \mathcal{T}_{\tilde{\pi}}Q_2\|_\infty \leq \gamma\|Q_1 - Q_2\|_\infty$. By the Banach fixed-point theorem, this operator has a unique fixed point $Q^{\tilde{\pi}}$. $\qquad\square$

**Theorem B.3** (Convergence and Asymptotic Performance Bound). *Under Assumptions B.2–B.5, the Q-learning process converges to the unique fixed point $Q^{\tilde{\pi}}$ of the modified operator $\mathcal{T}_{\tilde{\pi}}$ (or to its projected fixed point in the case of function approximation). The gap between this fixed point and the true soft Q-function $Q^{\pi_\theta}$ of policy $\pi_\theta$ under the original reward $r$ is bounded as follows:*

$$\|Q^{\pi_\theta} - Q^{\tilde{\pi}}\|_\infty \leq \frac{1}{1 - \gamma}\|(\mathcal{T}^{\pi_\theta} - \mathcal{T}_{\tilde{\pi}})Q^{\tilde{\pi}}\|_\infty$$

*Proof.* **Step 1: Convergence.** Theorem B.2 proves that $\mathcal{T}_{\tilde{\pi}}$ is a contraction mapping, thus guaranteeing a unique fixed point. When using function approximation, the Q-learning update can be viewed as finding the projected fixed point of the empirical operator $\hat{\mathcal{T}}_{\tilde{\pi}}$. Assumption B.5 ensures that the empirical operator converges uniformly to the true operator. Combined with standard theory for Fitted Q-Iteration (FQI), it can be shown that the learned Q-function $\hat{Q}_n$ converges to a neighborhood of the fixed point $Q^{\tilde{\pi}}$.

**Step 2: Performance Gap Analysis.** Our goal is to bound $\|Q^{\pi_\theta} - Q^{\tilde{\pi}}\|_\infty$. We know that $Q^{\pi_\theta}$ is the fixed point of the operator $\mathcal{T}^{\pi_\theta}$, i.e., $Q^{\pi_\theta} = \mathcal{T}^{\pi_\theta}Q^{\pi_\theta}$. We start from the target expression, use the triangle inequality, and decompose the error by adding and subtracting $\mathcal{T}^{\pi_\theta}Q^{\tilde{\pi}}$:

$$\|Q^{\pi_\theta} - Q^{\tilde{\pi}}\|_\infty = \|\mathcal{T}^{\pi_\theta}Q^{\pi_\theta} - Q^{\tilde{\pi}}\|_\infty$$

$$\leq \|\mathcal{T}^{\pi_\theta}Q^{\pi_\theta} - \mathcal{T}^{\pi_\theta}Q^{\tilde{\pi}}\|_\infty + \|\mathcal{T}^{\pi_\theta}Q^{\tilde{\pi}} - Q^{\tilde{\pi}}\|_\infty$$

For the first term, since $\mathcal{T}^{\pi_\theta}$ is also a $\gamma$-contraction mapping (the proof is analogous to that of Theorem B.2), we have:

$$\|\mathcal{T}^{\pi_\theta}Q^{\pi_\theta} - \mathcal{T}^{\pi_\theta}Q^{\tilde{\pi}}\|_\infty \leq \gamma\|Q^{\pi_\theta} - Q^{\tilde{\pi}}\|_\infty$$

For the second term, we use the fact that $Q^{\tilde{\pi}}$ is the fixed point of $\mathcal{T}_{\tilde{\pi}}$, i.e., $Q^{\tilde{\pi}} = \mathcal{T}_{\tilde{\pi}}Q^{\tilde{\pi}}$, and substitute it:

$$\|\mathcal{T}^{\pi_\theta}Q^{\tilde{\pi}} - Q^{\tilde{\pi}}\|_\infty = \|\mathcal{T}^{\pi_\theta}Q^{\tilde{\pi}} - \mathcal{T}_{\tilde{\pi}}Q^{\tilde{\pi}}\|_\infty = \|(\mathcal{T}^{\pi_\theta} - \mathcal{T}_{\tilde{\pi}})Q^{\tilde{\pi}}\|_\infty$$

Combining these results back into the main inequality:

$$\|Q^{\pi_\theta} - Q^{\tilde{\pi}}\|_\infty \leq \gamma\|Q^{\pi_\theta} - Q^{\tilde{\pi}}\|_\infty + \|(\mathcal{T}^{\pi_\theta} - \mathcal{T}_{\tilde{\pi}})Q^{\tilde{\pi}}\|_\infty$$

Finally, moving the $\|Q^{\pi_\theta} - Q^{\tilde{\pi}}\|_\infty$ term to the left-hand side and rearranging yields the final bound. $\qquad\square$

## C  DATASET SPECIFICATIONS

### C.1  SYNTHETIC TUMOR SIMULATION ENVIRONMENT

To facilitate the development and evaluation of dynamic treatment regimes, we designed a semi-synthetic data generation process. This process is inspired by the Tumor Growth (TG) simulator detailed in Geng et al. (2017), which models the longitudinal progression of tumor volume. Our simulation environment generates single-dimensional outputs representing tumor size and incorporates two primary therapeutic interventions: radiotherapy ($\mathbf{A}_t^r$) and chemotherapy ($\mathbf{A}_t^c$). A key modification in our work is the abstraction of these interventions as continuous values on the interval.

The core of the simulation is a dynamical system where tumor volume at time $t+1$ is a function of its prior state and the applied treatments. The interventions are characterized by distinct temporal dynamics: radiotherapy exhibits an immediate effect, $d(t)$, while chemotherapy has a cumulative and prolonged influence, $C(t)$. The mathematical formalization of this process is given by:

$$\mathbf{Y}_{t+1} = \left(1 + \rho \log\left(\frac{K}{\mathbf{Y}_t}\right) - \beta_C C(t) - (\alpha_r d(t) + \beta_r d(t)^2) + \epsilon_t\right) \mathbf{Y}_t, \tag{11}$$

where the parameters $\rho$ and $K$ govern the natural growth dynamics, and $\epsilon_t$ represents Gaussian noise drawn from $N(0, 0.01^2)$. To capture a more realistic, nonlinear dose-response relationship, the direct effects of the interventions, $d(t)$ and $C(t)$, are modeled via cubic spline transformations ($\psi_r$ and $\psi_c$) of the raw treatment assignments:

$$d(t) = 2\psi_r(\mathbf{A}_t^r), \tag{12}$$
$$C(t) = 5\psi_c(\mathbf{A}_t^c), \tag{13}$$

To emulate the heterogeneity observed in clinical populations, patient-specific responses to treatment are varied. This is achieved by sampling the response parameters $\beta_C, \alpha_r, \beta_r$ from a three-component truncated normal mixture distribution, where each component represents a latent patient subtype with fixed characteristics. Further details on parameterization are available in the accompanying anonymous repository.

A critical feature of this simulation is the incorporation of time-varying confounding, where treatment decisions are influenced by the patient's history. This is implemented through a biased assignment protocol, where the probability of receiving a given treatment is dependent on the recent trajectory of tumor growth. Both treatment assignments are drawn from a Beta distribution:

$$\mathbf{A}_t^r, \mathbf{A}_t^c \sim \text{Beta}(2\sigma_t, 2 - 2\sigma_t), \tag{14}$$

The shape of this distribution is dynamically adjusted by $\sigma_t$, which is calculated as follows:

$$\sigma_t = \sigma\left(\frac{\kappa}{D_{max}}\left(\bar{D}_{15}(\bar{\mathbf{Y}}_{t-1}) - D_{max}/2\right)\right), \tag{15}$$

Here, $\sigma(\cdot)$ is a sigmoid function, $\bar{D}_{15}(\bar{\mathbf{Y}}_{t-1})$ is the average tumor dimension over the last 15 days, and $D_{max}$ is the maximum tumor size. The parameter $\kappa$ explicitly controls the strength of this confounding effect; a $\kappa$ of zero results in random assignment, while larger values create a stronger dependency on patient history.

Finally, to assess the robustness of models to policy shifts, a distinct intervention strategy is introduced during the testing phase. With a probability of $\eta$ at each step, the standard treatment assignment is overridden by an independent policy where $\mathbf{A}_t^r, \mathbf{A}_t^c$ are drawn from a static $\text{Beta}(\alpha, \beta)$ distribution. The complete dataset comprises 1,000 trajectories for training, 100 for validation, and 100 for testing, with individual trajectories running for up to 60 time steps before termination due to patient outcomes. For evaluating model performance, we adopt the normalized target distance metric, consistent with established benchmarks (Bica et al., 2020; Melnychuk et al., 2022), calculated relative to a maximum tumor volume of $V_{\max} = 1150$ cm³.

## C.2 DETAILS ON EXPERIMENTS WITH SEMI-SYNTHETIC DATA

To simulate the complexity of real-world clinical data within a controlled environment, we designed a semi-synthetic data generation process. This process is built upon the MIMIC-III clinical database (Johnson et al., 2016) and leverages the standardized preprocessing pipeline from MIMIC-extract (Wang et al., 2020), which provides hourly aggregated ICU data. To ensure data quality, we imputed missing values in the time-series using a forward and backward filling strategy, and all continuous features were subsequently standardized. The intended clinical task in this environment is goal-conditioned control: drive a 2D outcome vector $(y_1, y_2)$ to a desired feasible target $\mathbf{Y}_{\text{target}}$, with a sparse goal-reaching reward (0 upon first hit, -1 otherwise) over a fixed planning horizon.

Our feature space is composed of 25 time-varying vital signs and 3 static covariates (gender, ethnicity, and age). To enable the model to process this categorical information, the static features were one-hot-encoded, resulting in a final 44-dimensional input vector ($d_v = 44$) for each timestep.

Our data generation process extends the methodology of Schulam & Saria (2017). The core principle is to first generate untreated outcome trajectories that evolve based on both endogenous dynamics and exogenous influences from patient covariates. After establishing these untreated paths, treatment effects are sequentially applied to construct the final trajectories. The model assumes a sparse dependency structure, meaning that an outcome is influenced by a limited number of covariates and treatments, and similarly, a treatment decision is informed by a limited set of factors.

**Cohort construction and sampling**. We adopted a semi-synthetic benchmark grounded in realistic clinical dynamics to evaluate our method, strictly following the experimental protocol established by Melnychuk et al. (2022). The study cohort was constructed based on the MIMIC-III database, from which we extracted hourly averaged physiological measurements of adult patients as the basis for simulation. To ensure a standardized evaluation environment, we implemented the following precise inclusion criteria: (1) we excluded all records with a length of stay shorter than 60 hours to ensure that the model has sufficient historical information for autoregressive modeling and to reduce padding artifacts; (2) using a fixed random seed (Seed=10), we drew a balanced cohort of $N = 500$ independent patient trajectories to ensure reproducibility; (3) all trajectories were strictly truncated to a fixed length of 60 hours, and missing values were imputed using forward and backward filling. The resulting state space contains 25 real-valued dynamic vital signs (e.g., heart rate, blood glucose) and 3 static demographic features. Based on these real physiological histories, we simulated synthetic patient outcomes and treatment effects to provide ground-truth benchmarks for causal inference. Notably, we extend the clinical abstraction of the original benchmark: while Melnychuk et al. (2022) simulate binary treatment decisions (i.e., presence or absence of an intervention), our framework models continuous interventions. This setting yields a more challenging control task, requiring the agent to determine precise continuous dosage levels in order to regulate synthetic health states driven by complex, realistic physiological dependencies.

The simulator operates through the following steps:

*First*, the process begins with the construction of a patient cohort of $n$ individuals, sampled randomly from ICU stays lasting at least 20 hours. We enforce fixed minimum/maximum sequence lengths and use a fixed seed for reproducibility; splits into train/val/test are also seed-controlled.

*Second*, we generate $d_y$ "untreated" outcome trajectories, $\mathbf{Z}t^{j,(i)}$, for each patient. These trajectories are a composite of three distinct sources: an endogenous component modeling patient-specific trends, an exogenous component capturing dependencies on covariates, and a stochastic noise term. The formal definition is:

$$\mathbf{Z}_t^{j,(i)} = \underbrace{\alpha_S^j \text{B-spline}(t) + \alpha_g^j g^{j,(i)}(t)}_{\text{endogenous}} + \underbrace{\alpha f^j f_Z^j \left( \mathbf{X}t^{(i)} \right)}_{\text{exogenous}} + \underbrace{\varepsilon_t}_{\text{noise}} \tag{16}$$

where the noise $\varepsilon t$ is drawn from $N(0, 0.005^2)$, and $\alpha_S^j$, $\alpha_g^j$, $\alpha_f^j$ serve as weighting parameters. The B-spline$(t)$ component is drawn from a mixture of three cubic splines, while the patient-specific function $g^{j,(i)}(\cdot)$ is drawn from a Gaussian Process with a Matérn kernel. The covariate-dependent function $f_Z^j(\cdot)$ is approximated using random Fourier features (RFF).

*Third*, we sequentially simulate $d_a$ continuous treatments $\mathbf{A}_t^l$ on the interval $(0,1)$. The assignment of these treatments is confounded by both a subset of current time-varying covariates (through the random function $f_Y^l(\mathbf{X}_t)$) and the historical average of previously treated outcomes over a window $T_l$ ($\bar{A}_{T_l}(\bar{\mathbf{Y}}_{t-1})$). These factors are integrated within a sigmoid function to yield a base probability, $pt^l$. This probability, along with a concentration parameter $c$, then defines a Beta distribution from which the final continuous treatment value, $\mathbf{A}_t^l$, is drawn. The process is formalized as:

$$p_t^l = \sigma \left( \Delta_A^l \bar{A}_{Tl} \left( \bar{\mathbf{Y}}_{t-1} \right) + \Delta_X^l f_Y^l \left( \mathbf{X}_t \right) + b_l \right) \quad \mathbf{A}_t^l \quad \sim \text{Beta} \left( c \cdot p_t^l, c \cdot (1 - p_t^l) \right) \tag{17}$$

where $\sigma(\cdot)$ is the sigmoid activation, $\Delta_A^l$ and $\Delta_X^l$ are confounding parameters, $b_l$ is a fixed bias, $c$ is the base concentration parameter, and $f_Y^l(\cdot)$ is sampled from an RFF approximation of a Gaussian process.

*Fourth*, we apply treatments to the untreated outcomes, initializing with $\mathbf{Y}_1 = \mathbf{Z}_1$. Each treatment is modeled to have a lasting influence on specific outcomes. The maximal additive effect is determined by transforming the sampled treatment value using a cubic spline function, denoted $\text{cs}(\cdot)$. This effect is applied over a time window $t - w^l, \ldots, t - 1$ and is subject to an inverse-square-root decay. When multiple treatments are active, their combined influence is determined by taking the minimum effect at each time step. The total effect $E^j(t)$ is modeled as:

$$E_{(}^j t) = \sum_{i=t-w^l}^{t-1} \frac{\min_{l=1,\ldots,d_a} \left( \text{cs}(\mathbf{A}_i^l) \cdot \beta_{lj} \right)}{\sqrt{t-i}}, \tag{18}$$

where $\beta_{lj}$ represents the maximum effect size of treatment $l$ on outcome $j$.

*Fifth*, the final outcome at each timestep is then synthesized by adding the aggregated treatment effect $E^j(t)$ to the untreated trajectory:

$$\mathbf{Y}_t^j = \mathbf{Z}_t^j + E^j(t). \tag{19}$$

*Sixth*, the output of this simulation process is our final semi-synthetic dataset. Based on the generation of three continuous treatments ($d_a = 3$) and two outcomes ($d_y = 2$), the patient cohort is partitioned into training, validation, and testing sets. In summary, the designed target variables are the 2D outcome vector $(y_1, y_2)$ and the task is to reach a specified clinically plausible target region. Cohort selection is seeded and criteria-based; evaluation focuses on goal-conditioned planning performance under controlled yet physiologically grounded dynamics, rather than disease-specific effect estimation.

## D HYPERPARAMETER TUNING

The hyperparameter settings for our proposed GIFT model are detailed in Table 4. We employed a random grid search methodology to optimize these parameters. For all baseline models, including RMSN, CRN, CT, ACTIN and VCIP, we adopted the hyperparameter optimization strategy and search ranges consistent with those reported in their original studies. This ensures a fair and robust comparison across all evaluated methods.

Table 4: Specified ranges for hyperparameter tuning of GIFT across various datasets.

| Hyperparameter | Range (tumor) | Range (MIMIC-III) |
|---|---|---|
| Learning rate (SAC) $l$ | 5e-4, 1e-3, 2e-3 | 5e-4, 1e-3, 2e-3 |
| Minibatch size | 128, 256, 512, 1024 | 128, 256, 512, 1024 |
| Hidden size | 32, 72, 112 | 32, 72, 112 |
| SAC Actor/Critic hidden layers | [32], [64], [128] | [32], [64], [128] |
| History Encoder hidden layers | [32], [64], [128] | [32], [64], [128] |
| Discount factor $\gamma$ | 0.5, 0.7, 0.9 | 0.5, 0.7, 0.9 |
| HER future goals $k$ | 0, 3, 5 | 0, 3, 5 |
| $\epsilon_1, \epsilon_2$ | 0.01,10 | 0.01,10 |
| Target hit ratio | [0.45, 0.5, ..., 0.9] | [0.25, 0.3, ..., 0.75] |
| Number of epochs | 15 | 30 |

## E UTILIZATION OF LARGE LANGUAGE MODELS

Large Language Models (LLMs) were strategically employed throughout this research to enhance productivity and output quality. In the writing process, LLMs were utilized for text refinement and polishing, improving the clarity, coherence, and academic tone of the manuscript while ensuring consistent writing style and precise terminology. For data analysis, LLMs generated comprehensive analytical code including data preprocessing routines, statistical analysis functions, and visualization scripts, which accelerated the experimental result interpretation workflow. Additionally, during the initial research phase, LLMs assisted in conducting literature surveys, synthesizing information from multiple sources, and identifying relevant research directions.

## F ADDITIONAL EXPERIMENTS

### F.1 ANALYSIS OF GOAL ACHIEVEMENT SPARSITY.

In observational data, trajectories successfully reaching specific goals can be exceedingly rare, posing significant challenges for policy learning. In our problem formulation (Section 3), the target region $\mathcal{T}$ is defined by radius $\delta$. The proportion of training trajectories within this region, termed "Hit Ratio," correlates with $\delta$ choice. This introduces a critical trade-off: smaller $\delta$ results in lower Hit Ratio, making reward signals sparse and hindering learning. Conversely, overly large $\delta$ renders goals too lenient, leading models to learn policies achieving high hit rates while remaining far from target $\mathbf{Y}_{target}$. Figure 7 illustrates model RMSE versus Hit Ratio on both datasets, validating this trade-off.

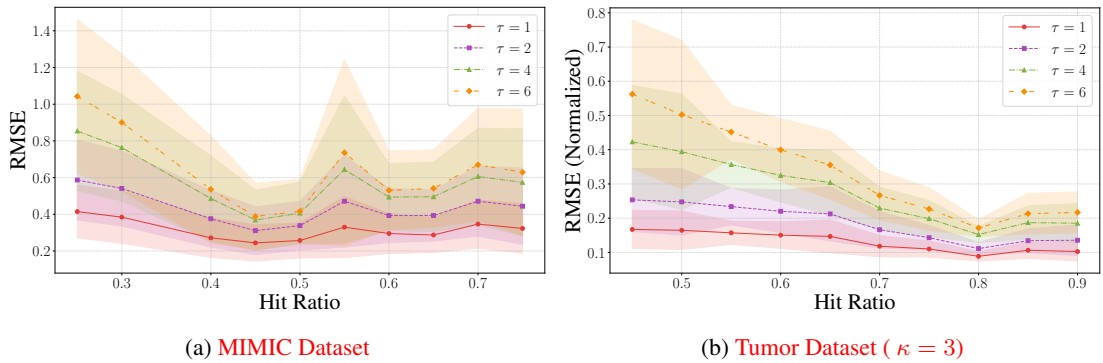

(a) MIMIC Dataset

(b) Tumor Dataset ( $\kappa = 3$ )

Figure 7: Performance evaluation of our model, presented as RMSE, on the (a) MIMIC and (b) Tumor datasets. The comparison is made across varying Hit Ratios (the proportion of training data reaching a target threshold) and for different values of the hyperparameter $\tau$.

## F.2 ANALYSIS OF OPTIMIZATION STRATEGIES

To investigate the impact of different optimization strategies, we compare our proposed model, GIFT, with baselines trained under two distinct paradigms: step-level and episode-level optimization. The results on the MIMIC-III and Tumor datasets are presented in Tables 5 through 8.

**Step-level vs. Episode-level Optimization.** For all baseline models (RMSN, CRN, CT, ACTIN, and VCIP), a clear and consistent trend emerges across all datasets: episode-level optimization consistently outperforms step-level optimization. The performance gap is particularly pronounced for longer prediction horizons ($\tau$). For instance, in Table 5, the error for RMSN under the step-level strategy escalates to $2.13 \pm 0.57$ at $\tau = 6$, whereas the episode-level strategy maintains a much lower error of $0.83 \pm 0.28$. This suggests that optimizing over the entire sequence trajectory (episode-level) is more effective for long-term forecasting than the myopic, step-by-step approach, which is susceptible to the compounding of errors over time.

**Superiority of GIFT.** Our proposed model, GIFT, significantly surpasses all baseline models, regardless of their optimization strategy. As evidenced by the bolded results in the tables, GIFT achieves the lowest error across all prediction horizons and datasets. This performance advantage is not only substantial but also grows as the prediction horizon $\tau$ increases. For example, on the Tumor dataset with $\kappa = 4$ (Table 8), GIFT's error at $\tau = 6$ is $0.40 \pm 0.05$, which is less than half that of the best-performing baseline, VCIP ($0.87 \pm 0.26$). This demonstrates that GIFT's inherent architecture and training mechanism provide a more robust solution, effectively mitigating the challenges of long-term sequential forecasting without being constrained by the choice between step-level and episode-level optimization. The results firmly establish the superiority of our proposed approach.

## F.3 IMPACT OF INFERENCE OPTIMIZATION STEPS

To investigate the impact of inference-time optimization, we analyzed model performance as a function of optimization steps on both the MIMIC and Tumor datasets (Figures 8-12). The results reveal that the optimal optimization strategy is highly dependent on the dataset's characteristics.

On the MIMIC dataset, most models (e.g., RMSN, CRN, ACTIN) exhibit robust behavior: the RMSE decreases with optimization and then converges to a stable plateau. This indicates that after reaching a certain performance level, additional optimization steps do not degrade performance.

Table 5: Performance comparison of GIFT and baseline models under various optimization strategies on the MIMIC-III dataset.

| Model | Strategy | $\tau = 1$ | $\tau = 2$ | $\tau = 3$ | $\tau = 4$ | $\tau = 5$ | $\tau = 6$ |
|---|---|---|---|---|---|---|---|
| RMSN | step | 0.26±0.06 | 0.61±0.18 | 1.02±0.28 | 1.42±0.40 | 1.76±0.45 | 2.13±0.57 |
| RMSN | episode | 0.25±0.07 | 0.39±0.13 | 0.50±0.17 | 0.60±0.20 | 0.70±0.21 | 0.83±0.28 |
| CRN | step | 0.43±0.04 | 0.78±0.13 | 1.12±0.30 | 1.49±0.46 | 1.80±0.58 | 2.14±0.73 |
| CRN | episode | 0.31±0.04 | 0.46±0.11 | 0.60±0.15 | 0.71±0.17 | 0.83±0.18 | 0.96±0.24 |
| CT | step | 0.92±0.31 | 1.52±0.62 | 2.01±0.89 | 2.44±1.14 | 2.86±1.35 | 3.23±1.55 |
| CT | episode | 0.62±0.17 | 1.06±0.32 | 1.42±0.47 | 1.72±0.60 | 1.98±0.73 | 2.20±0.83 |
| ACTIN | step | **0.24±0.20** | 0.62±0.38 | 1.12±0.51 | 1.56±0.60 | 2.01±0.74 | 2.45±0.85 |
| ACTIN | episode | 0.28±0.19 | 0.57±0.34 | 0.79±0.50 | 0.98±0.64 | 1.11±0.74 | 1.23±0.81 |
| VCIP | step | 0.41±0.19 | 0.54±0.24 | 0.60±0.25 | 0.63±0.26 | 0.66±0.26 | 0.68±0.25 |
| VCIP | episode | 0.41±0.19 | 0.51±0.22 | 0.57±0.21 | 0.60±0.20 | 0.62±0.18 | 0.67±0.22 |
| GIFT | | 0.24±0.10 | **0.31±0.13** | **0.33±0.14** | **0.37±0.16** | **0.38±0.17** | **0.39±0.18** |

Table 6: Performance comparison of GIFT and baseline models under various optimization strategies on the Tumor dataset ($\kappa = 2$).

| Model | Strategy | $\tau = 1$ | $\tau = 2$ | $\tau = 3$ | $\tau = 4$ | $\tau = 5$ | $\tau = 6$ |
|---|---|---|---|---|---|---|---|
| RMSN | step | 0.15±0.05 | 0.27±0.09 | 0.36±0.11 | 0.44±0.15 | 0.51±0.19 | 0.58±0.24 |
| RMSN | episode | 0.09±0.03 | 0.14±0.05 | 0.18±0.06 | 0.21±0.07 | 0.25±0.09 | 0.28±0.12 |
| CRN | step | 0.16±0.05 | 0.27±0.10 | 0.37±0.12 | 0.44±0.16 | 0.50±0.19 | 0.56±0.24 |
| CRN | episode | 0.10±0.03 | 0.15±0.06 | 0.19±0.07 | 0.23±0.08 | 0.26±0.10 | 0.30±0.12 |
| CT | step | 0.17±0.04 | 0.29±0.09 | 0.38±0.12 | 0.45±0.15 | 0.51±0.18 | 0.57±0.23 |
| CT | episode | 0.11±0.03 | 0.17±0.06 | 0.21±0.07 | 0.25±0.09 | 0.29±0.11 | 0.34±0.16 |
| ACTIN | step | 0.18±0.06 | 0.32±0.14 | 0.41±0.20 | 0.47±0.24 | 0.53±0.28 | 0.59±0.32 |
| ACTIN | episode | 0.14±0.06 | 0.22±0.09 | 0.29±0.11 | 0.34±0.13 | 0.41±0.15 | 0.46±0.19 |
| VCIP | step | 0.10±0.03 | 0.15±0.05 | 0.19±0.06 | 0.22±0.08 | 0.26±0.10 | 0.29±0.12 |
| VCIP | episode | 0.10±0.03 | 0.16±0.05 | 0.18±0.05 | 0.21±0.07 | 0.25±0.10 | 0.28±0.13 |
| GIFT | | **0.08±0.02** | **0.10±0.03** | **0.11±0.04** | **0.12±0.04** | **0.13±0.05** | **0.14±0.05** |

In stark contrast, on the Tumor dataset, the dominant trend for most models (RMSN, CRN, VCIP) is a distinct **U-shaped performance curve**. The RMSE initially improves but then degrades with excessive optimization, highlighting a significant risk of overfitting. This makes the precise number of optimization steps a critical hyperparameter for this dataset.

A consistent finding across both datasets is the instability of the CT model, which performs poorly with increased optimization, especially for long-term prediction. Conversely, the ACTIN model consistently showed the most robust behavior on both datasets. In conclusion, while inference-time optimization is a powerful technique, its application requires careful tuning tailored to the specific model and dataset to avoid potential overfitting.

Table 7: Performance comparison of GIFT and baseline models under various optimization strategies on the Tumor dataset ($\kappa = 3$).

| Model | Strategy | $\tau = 1$ | $\tau = 2$ | $\tau = 3$ | $\tau = 4$ | $\tau = 5$ | $\tau = 6$ |
|---|---|---|---|---|---|---|---|
| RMSN | step | 0.23±0.04 | 0.40±0.06 | 0.57±0.08 | 0.73±0.12 | 0.86±0.17 | 0.99±0.24 |
| RMSN | episode | 0.12±0.02 | 0.21±0.03 | 0.29±0.06 | 0.35±0.10 | 0.41±0.12 | 0.46±0.14 |
| CRN | step | 0.33±0.06 | 0.55±0.08 | 0.75±0.13 | 0.89±0.15 | 1.01±0.22 | 1.10±0.24 |
| CRN | episode | 0.18±0.05 | 0.28±0.07 | 0.39±0.10 | 0.48±0.11 | 0.56±0.15 | 0.63±0.16 |
| CT | step | 0.34±0.08 | 0.59±0.12 | 0.80±0.19 | 0.94±0.21 | 1.07±0.28 | 1.15±0.30 |
| CT | episode | 0.21±0.06 | 0.33±0.05 | 0.45±0.08 | 0.55±0.09 | 0.65±0.13 | 0.72±0.13 |
| ACTIN | step | 0.29±0.04 | 0.45±0.09 | 0.59±0.09 | 0.68±0.10 | 0.76±0.11 | 0.84±0.14 |
| ACTIN | episode | 0.29±0.04 | 0.39±0.03 | 0.51±0.03 | 0.61±0.04 | 0.71±0.07 | 0.78±0.09 |
| VCIP | step | 0.16±0.04 | 0.23±0.04 | 0.29±0.06 | 0.33±0.09 | 0.39±0.11 | 0.40±0.08 |
| VCIP | episode | 0.16±0.04 | 0.22±0.04 | 0.29±0.07 | 0.33±0.08 | 0.38±0.10 | 0.41±0.09 |
| GIFT | | **0.11±0.02** | **0.14±0.04** | **0.17±0.04** | **0.20±0.05** | **0.21±0.06** | **0.23±0.06** |

Table 8: Performance comparison of GIFT and baseline models under various optimization strategies on the Tumor dataset ($\kappa = 4$).

| Model | Strategy | $\tau = 1$ | $\tau = 2$ | $\tau = 3$ | $\tau = 4$ | $\tau = 5$ | $\tau = 6$ |
|---|---|---|---|---|---|---|---|
| RMSN | step | 0.59±0.09 | 1.04±0.19 | 1.39±0.18 | 1.69±0.19 | 1.93±0.20 | 2.10±0.22 |
| RMSN | episode | 0.29±0.05 | 0.45±0.08 | 0.61±0.13 | 0.75±0.16 | 0.91±0.20 | 1.01±0.21 |
| CRN | step | 0.62±0.20 | 1.09±0.27 | 1.42±0.30 | 1.70±0.30 | 1.94±0.28 | 2.11±0.27 |
| CRN | episode | 0.34±0.14 | 0.59±0.26 | 0.77±0.37 | 0.92±0.45 | 1.07±0.50 | 1.14±0.54 |
| CT | step | 0.66±0.18 | 1.13±0.28 | 1.52±0.34 | 1.80±0.33 | 2.02±0.36 | 2.16±0.32 |
| CT | episode | 0.40±0.16 | 0.68±0.26 | 0.98±0.36 | 1.20±0.40 | 1.40±0.41 | 1.57±0.44 |
| ACTIN | step | 0.50±0.16 | 0.87±0.31 | 1.13±0.37 | 1.34±0.38 | 1.55±0.41 | 1.70±0.38 |
| ACTIN | episode | 0.50±0.15 | 0.85±0.29 | 1.08±0.35 | 1.27±0.39 | 1.45±0.44 | 1.60±0.50 |
| VCIP | step | 0.34±0.07 | 0.48±0.12 | 0.60±0.18 | 0.69±0.19 | 0.78±0.23 | 0.87±0.26 |
| VCIP | episode | 0.34±0.07 | 0.46±0.11 | 0.58±0.18 | 0.69±0.19 | 0.81±0.23 | 0.87±0.24 |
| GIFT | | **0.25±0.05** | **0.32±0.08** | **0.35±0.05** | **0.38±0.04** | **0.39±0.04** | **0.40±0.05** |

# G  BASELINE DETAILS

## G.1  STABLE CONTRASTIVE RL (SCRL) FOR SCTA

We adopt Stable Contrastive RL (SCRL) (Zheng et al., 2024) as a general offline GCRL baseline and instantiate it for the SCTA task. The core idea is to construct a discriminative reachability score between "history–action" and "goal," and use this score to directly drive goal-conditioned policy learning, thereby obtaining a closed-loop decision maker from offline data.

Concretely, a variable-length history (past outcomes, interventions, static features, and optional vitals) is encoded into a fixed-dimensional history representation $z_h$, and the target outcome vector is mapped to a

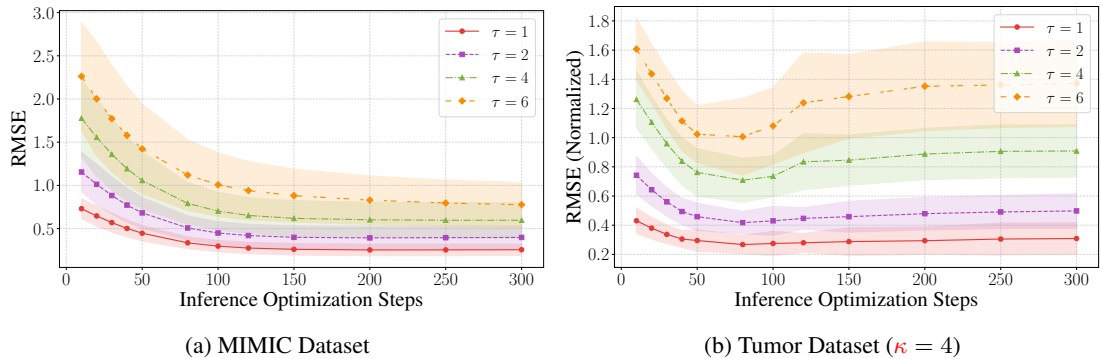

(a) MIMIC Dataset

(b) Tumor Dataset ($\kappa = 4$)

Figure 8: Performance of the RMSN model with the episode-level optimization strategy as a function of the number of inference optimization steps. The plots show the RMSE for different forecast horizons ($\tau \in \{1, 2, 4, 6\}$) on (a) the MIMIC dataset and (b) the Tumor dataset ($\kappa = 4$).

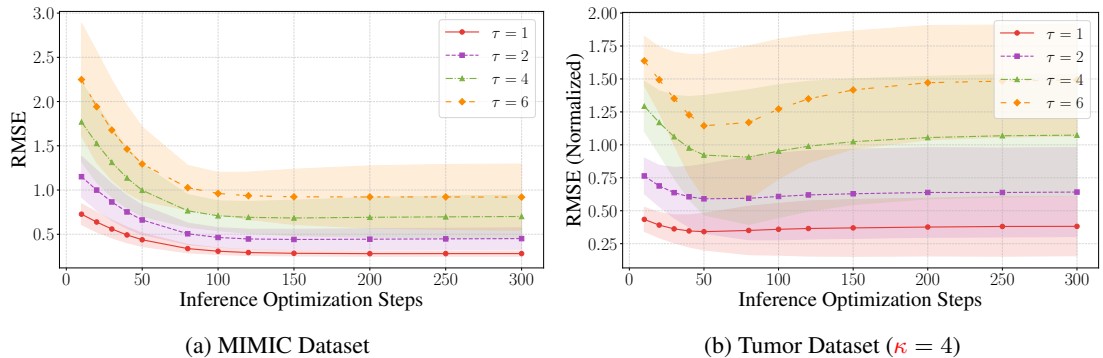

(a) MIMIC Dataset

(b) Tumor Dataset ($\kappa = 4$)

Figure 9: Performance of the CRN model with the episode-level optimization strategy as a function of the number of inference optimization steps. The plots show the RMSE for different forecast horizons ($\tau \in \{1, 2, 4, 6\}$) on (a) the MIMIC dataset and (b) the Tumor dataset ($\kappa = 4$).

goal representation $z_g$. Two mappings, $\phi(s, a)$ and $\psi(g)$, project "history–action" and "goal" into a shared embedding space, where normalized similarity (e.g., cosine) measures reachability:

$$\text{sim}(\phi, \psi) = \left\langle \frac{\phi}{\|\phi\|}, \ \frac{\psi}{\|\psi\|} \right\rangle.$$

The critic is trained with temperature-scaled InfoNCE/cross-entropy (positives are matched $(\phi, \psi)$ pairs; negatives come from in-batch pairing and a queue-based memory bank):

$$\mathcal{L}_{\text{critic}} = \text{CE}\left( \frac{\phi \, \psi^\top}{\eta} \right),$$

where $\eta$ is the temperature. The policy takes $(z_h, z_g)$ as input and outputs continuous interventions, maximizing the similarity between policy-induced actions and the goal embedding while adding a behavior cloning (BC) regularizer to suppress out-of-distribution actions in the offline setting:

$$\mathcal{L}_{\text{actor-ctr}} = - \mathbb{E}\big[\text{sim}\big(\phi(z_h, \pi(z_h, z_g)), \ \psi(z_g)\big)\big],$$

$$\mathcal{L}_{\text{actor}} = \mathcal{L}_{\text{actor-ctr}} + \lambda_{\text{BC}} \cdot \big( - \mathbb{E}[\log \pi(a_{\text{data}} \mid z_h, z_g)]\big).$$

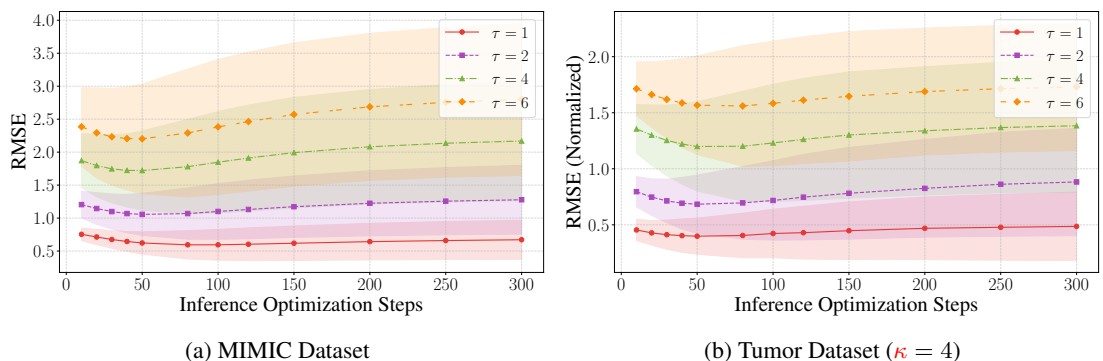

(a) MIMIC Dataset

(b) Tumor Dataset ($\kappa = 4$)

Figure 10: Performance of the CT model with the episode-level optimization strategy as a function of the number of inference optimization steps. The plots show the RMSE for different forecast horizons ($\tau \in \{1, 2, 4, 6\}$) on (a) the MIMIC dataset and (b) the Tumor dataset ($\kappa = 4$).

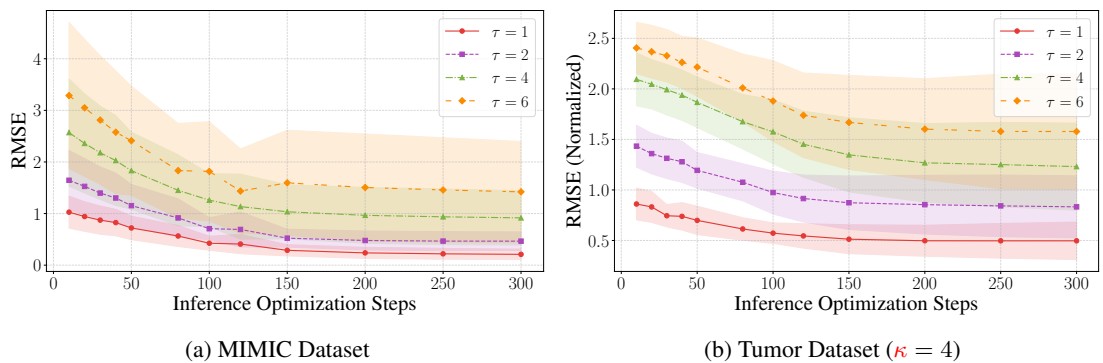

(a) MIMIC Dataset

(b) Tumor Dataset ($\kappa = 4$)

Figure 11: Performance of the ACTIN model with the episode-level optimization strategy as a function of the number of inference optimization steps. The plots show the RMSE for different forecast horizons ($\tau \in \{1, 2, 4, 6\}$) on (a) the MIMIC dataset and (b) the Tumor dataset ($\kappa = 4$).

To improve stability and generalization, following Zheng et al. (2024), we use large-batch training, layer normalization, lightweight data augmentation, and small-range cold-start initialization.

Training proceeds as purely offline alternating optimization: first update the critic (minimize $\mathcal{L}_{\text{critic}}$ and maintain the negative sample queue), then update the policy (minimize $\mathcal{L}_{\text{actor}}$). At inference time, given "history and goal," the policy outputs actions in a closed-loop manner, rolling the history forward with environment/simulator feedback until reaching the goal or the planning horizon. Compared with open-loop plans that rely on test-time search, the closed-loop policy has lower inference latency.

Regarding empirical performance, the results reported in Tables 1 and 2 show that, on the tumor dataset ($\kappa=4$, different intervention strategies between train and test) and the semi-synthetic MIMIC dataset (same strategy between train and test), SCRL attains higher terminal RMSE than GIFT, with the gap widening as the planning horizon $\tau$ increases. A plausible explanation is that, in TCTA, the goal is a low-dimensional clinical target rather than a "future observation," so contrastive alignment tends to degenerate into static similarity and struggles to capture the temporal signal of "dynamical reachability." In addition, SCTA requires

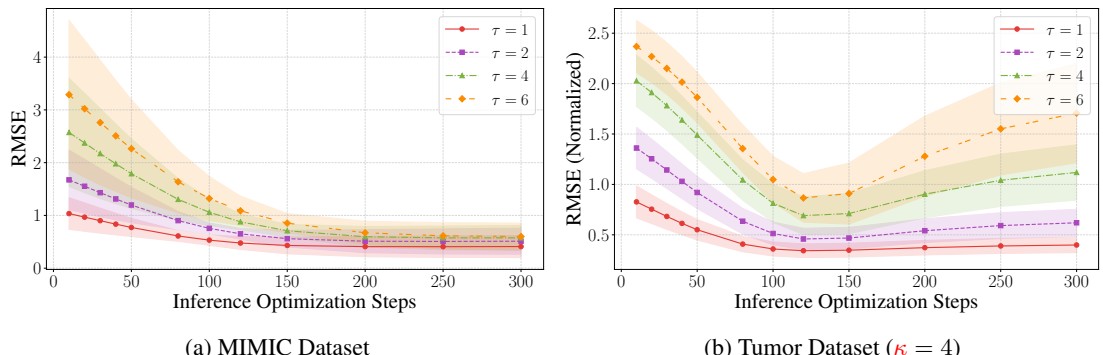

(a) MIMIC Dataset

(b) Tumor Dataset ($\kappa = 4$)

Figure 12: Performance of the VCIP model with the episode-level optimization strategy as a function of the number of inference optimization steps. The plots show the RMSE for different forecast horizons ($\tau \in \{1, 2, 4, 6\}$) on (a) the MIMIC dataset and (b) the Tumor dataset ($\kappa = 4$).

handling both causal identifiability and robust estimation under offline distribution shift ($\pi_b \rightarrow \pi_\theta$), which goes beyond the typical assumptions of general GCRL.

## G.2 DEFINITIONS OF OPTIMIZATION STRATEGIES

As mentioned in the main text, for baseline models incapable of directly outputting decisions, we devised two optimization strategies to find the optimal treatment sequence during the inference phase. This section provides a detailed description of their definitions, objective functions, and algorithmic implementations.

**Step-level Optimization** is an online, greedy approach. At each decision-making time step, the model employs an iterative optimization process (as shown in Algorithm 1) to determine the optimal action for the current step. The objective is to identify the action that brings the subsequent prediction closest to the final target.

**Episode-level Optimization** is an offline, global method. This strategy treats all future $\tau$ treatment actions as a complete sequence and jointly optimizes this entire sequence via gradient descent (as detailed in Algorithm 2). The goal is to discover the full sequence of actions that minimizes the discrepancy between the final predicted outcome and the target.

## G.3 OBJECTIVE FUNCTIONS

The core of both optimization strategies is the minimization of an objective function, `calculate_objective`, whose specific definition varies depending on the model paradigm.

For **VCIP**, this function computes the negative of the Evidence Lower Bound (ELBO). The optimization goal is to maximize the conditional likelihood of achieving the target, which is equivalent to minimizing the following objective function:

$$\mathcal{L}_{\text{VCIP}}(\overline{\mathbf{a}}_{t,\tau}) = -\text{ELBO}(Y_{\text{target}}|\overline{\mathbf{H}}_t, \overline{\mathbf{a}}_{t,\tau}) \tag{20}$$

For **other baselines** (RMSN, CRN, CT, and ACTIN), the function calculates the Mean Squared Error (MSE) between the predicted counterfactual outcome and the target. The objective is to directly minimize the distance between them:

$$\mathcal{L}_{\text{Baselines}}(\overline{\mathbf{a}}_{t,\tau}) = \|\hat{Y}[\overline{\mathbf{a}}_{t,\tau}] - Y_{\text{target}}\|_2^2 \tag{21}$$

## G.4 ALGORITHMS

---

**Algorithm 1** Step-level Optimization Strategy for Baselines

---

**Require:** Baseline model $M$, initial history $\bar{\mathbf{h}}_t$, target $\mathbf{y}_{\text{target}}$, number of decision steps $\tau$, real environment/simulator $\mathcal{E}$
**Require:** Optimization steps per action $K$, learning rate $\alpha$
1: Initialize the full treatment sequence $\bar{\mathbf{a}}_{t,\tau} \leftarrow \emptyset$
2: $\bar{\mathbf{h}}_{\text{current}} \leftarrow \bar{\mathbf{h}}_t$
3: **for** $k = 0, \ldots, \tau - 1$ **do**
4:     Randomly initialize action for the current step $\mathbf{a}_k$
5:     **for** step $= 1$ to $K$ **do**                     ▷ Find the optimal action for the current step
6:         $\mathcal{L} \leftarrow$ calculate_objective$(M, \bar{\mathbf{h}}_{\text{current}}, \mathbf{a}_k, \mathbf{y}_{\text{target}})$
7:         $\mathbf{a}_k \leftarrow \mathbf{a}_k - \alpha \nabla_{\mathbf{a}_k} \mathcal{L}$
8:     **end for**
9:     $\mathbf{a}^* \leftarrow \mathbf{a}_k$                                    ▷ Obtain the optimal action for the current step
10:    $\bar{\mathbf{a}}_{t,\tau} \leftarrow \bar{\mathbf{a}}_{t,\tau} \cup \{\mathbf{a}^*\}$
11:    $(\mathbf{x}_{\text{next}}, \mathbf{y}_{\text{next}}) \leftarrow \mathcal{E}.\text{step}(\mathbf{a}^*)$      ▷ Execute action and get the next state from the real system
12:    $\bar{\mathbf{h}}_{\text{current}} \leftarrow$ update_history$(\bar{\mathbf{h}}_{\text{current}}, \mathbf{a}^*, \mathbf{x}_{\text{next}}, \mathbf{y}_{\text{next}})$      ▷ Update history with real feedback
13: **end for**
14: **return** $\bar{\mathbf{a}}_{t,\tau}$

---

---

**Algorithm 2** Episode-level Optimization Strategy for Baselines

---

**Require:** Baseline model $M$, initial history $\bar{\mathbf{h}}_t$, target $\mathbf{y}_{\text{target}}$, number of decision steps $\tau$, learning rate $\alpha$, number of optimization steps $K$
1: Randomly initialize the treatment sequence $\bar{\mathbf{a}}_{t,\tau}$
2: **for** step $= 1$ to $K$ **do**
3:     $\mathcal{L} \leftarrow$ calculate_objective$(M, \bar{\mathbf{h}}_t, \bar{\mathbf{a}}_{t,\tau}, \mathbf{y}_{\text{target}})$          ▷ Calculate the global objective
4:     $\bar{\mathbf{a}}_{t,\tau} \leftarrow \bar{\mathbf{a}}_{t,\tau} - \alpha \nabla_{\bar{\mathbf{a}}_{t,\tau}} \mathcal{L}$      ▷ Update the entire treatment sequence via gradient descent
5: **end for**
6: **return** $\bar{\mathbf{a}}_{t,\tau}$

---

