# OpenReview forum: "Target-Driven Policy Optimization for Sequential Counterfactual Outcome Control"
_ICLR.cc/2026/Conference — Submitted to ICLR 2026_

### Official Review · Reviewer_equA · 2025-10-31

**Soundness:** 2
**Presentation:** 2
**Contribution:** 2
**Rating:** 4
**Confidence:** 3

**Summary:**

This paper addresses the challenge of learning sequential intervention policies from offline observational data to guide temporal systems toward achieving desired outcomes, which is particularly relevant in fields such as personalized medicine. The authors introduce GIFT (Goal-conditioned Intervention via Factual-Targeted Training), a framework that frames the problem as a goal-conditioned Markov Decision Process. It combines Hindsight Experience Replay (HER) with a clipped importance-weighted reward rescaling mechanism to tackle issues related to sparse rewards and distributional shifts in offline reinforcement learning. The paper presents a theoretical analysis demonstrating the convergence of a modified Bellman operator and provides empirical results from synthetic and semi-synthetic datasets, indicating that GIFT outperforms several baseline methods in terms of accuracy, generalization, and inference efficiency.

**Strengths:**

Overall, the paper is well organized and addresses a meaningful application. The problem setting presents practical challenges, and integrating HER with offline RL is a complex task. The experiments are thorough and support the effectiveness of the proposed GIFT method.

**Weaknesses:**

-  The core methodological contributions appear incremental. The use of HER for goal relabeling and clipped importance sampling for stabilizing off-policy learning are well-established techniques in the RL literature; the paper largely recombines these ideas without introducing a fundamentally new algorithmic principle. The theoretical results, though correctly derived, follow standard analyses of contraction mappings and bias-variance trade-offs in approximate dynamic programming, offering limited conceptual advancement. Moreover, the claimed novelty of framing counterfactual target achievement as a goal-conditioned MDP is more a reframing of existing dynamic treatment regimes or model-predictive control formulations than a paradigm shift.

- The experiments do not convincingly demonstrate that the observed gains stem from a novel insight. The ablation studies confirm the utility of HER and reward rescaling, but do not isolate a unique contribution beyond their combined effect. Additionally, the paper does not sufficiently differentiate itself from prior work in offline goal-conditioned RL or conservative policy learning (e.g., CQL), and the comparison with CQL is treated as an ablation rather than a serious alternative

**Questions:**

- How does this work differ from prior work in offline goal-conditioned RL?

---

> ### Author Response · Authors · 2025-11-17
> **Part I of the response**
>
> **Regarding "methodological contribution is incremental, mainly combining HER with clipped importance sampling, theory is routine, reframing the problem as GCRL/DTR/MPC":**
>
> The research objective of this paper is not to propose a general-purpose GCRL algorithm, but rather to answer whether, under realistic conditions with confounding and offline-only training, SCTA can be formulated as a "goal-conditioned MDP" that achieves causal identifiability on observational data and enables stable learning through a convergent operator. To this end, we fill two literature gaps: First, **regarding identifiability, Proposition 4.1** proves that under goal-independent dynamics and standard causal assumptions, the expectation of the soft Bellman operator can be directly expressed through one-step importance weighting of the **observational distribution**, thereby avoiding the high variance of long-trajectory ratios; Second, regarding offline stable learning, we apply bounded clipped ratio reweighting to immediate rewards $\tilde{r}(\psi,a)=\bar{\rho}(\psi,a) r(\psi,a)$, where $\rho(\psi,a)=\pi\_\theta(a\mid\psi)/\pi\_b(a\mid\psi)$ and $\bar{\rho}=\mathrm{clip}(\rho,\epsilon_1,\epsilon_2)$. The corresponding "modified operator" maintains $\gamma$-contraction (Theorem 4.2) and provides a closed-form performance upper bound equivalent to the clipping bias (Theorem 4.3). Additionally, we explicitly provide an **observationally identifiable form of the soft operator** that unifies "goal-conditioned evaluation" and "offline distribution shift correction" within the same Bellman framework. This differs from DTR's overall effect maximization and MPC/offline planning's open-loop search during inference: we directly learn a closed-loop goal-conditioned policy, avoiding inference-time optimization that is costly and sensitive to model errors. Both theory and algorithm address the offline and confounding bottlenecks of SCTA, not a simple juxtaposition of existing techniques.
>
> **Regarding "empirical results fail to demonstrate source of gains, insufficient distinction from CQL, CQL only used as ablation":**
>
> The gains primarily stem from the synergy of two mechanisms: HER relabels failed experiences as learnable "goal-hitting" samples to alleviate sparse rewards; reward reweighting uses single-step, clipped importance ratios to align the evaluation policy distribution and control variance. Ablation studies (Table 3) show that removing either component leads to significant degradation, and reward reweighting significantly reduces training variance (e.g., on MIMIC with $\tau=3$, standard deviation decreases from $0.39$ to $0.14$), consistent with our "single-step clipping to avoid trajectory-level variance" design. **Sensitivity analysis (Figure 6)** further reveals the bias-variance trade-off of $\epsilon_2$ and its stable working range, indicating the improvement is not due to fortuitous hyperparameter tuning. More critically, significant advantages are maintained in extrapolation evaluation on "unseen intervention policies" (Table 1), directly pointing to the effectiveness of offline distribution shift handling. Regarding the relationship with CQL, their objectives differ: CQL applies conservative penalties on OOD actions at the critic level and does not specifically address sparse reward alignment for "first-hitting goal sets"; we apply $\bar{\rho}$ reweighting to immediate rewards at the operator level, maintaining contraction while providing an interpretable upper bound, directly aligning the learning signal for goal achievement. Empirically, "with CQL" outperforms "w/o RR" but is systematically inferior to the complete GIFT (Table 3), supporting this distinction. To prevent readers from categorizing this work as "general offline GCRL," the revised version adds a carefully tuned **Stable Contrastive RL (SCRL) baseline [1]**. SCRL's empirical results are suboptimal (**Fig. 3, Tables 1 and 2**), with unsatisfactory performance: its assumption that "goal ≈ future observation" and contrastive temporal signals degenerate into static similarity learning under low-dimensional clinical goals, unable to express "reachability" and failing to drive effective goal-achievement policies.
>
> [1] Zheng et al., 2024. Stabilizing Contrastive RL: Techniques for Robotic Goal Reaching from Offline Data. In International Conference on Learning Representations (ICLR), 2024.

---

> ### Author Response · Authors · 2025-11-17
> **Part II of the response**
>
> **Regarding "how to distinguish from existing offline GCRL work":**
>
> We sincerely appreciate the reviewer's feedback. We recognize that the current writing makes it easier for readers to categorize this paper as "general offline GCRL," but this is not our research objective. We acknowledge that most reviewers have questions about this, and we have provided clarification in the **Consolidated Response and Clarification**, explaining the differences between this work and "general offline GCRL."
>
> Existing offline GCRL methods mostly operate in confounding-free visual/robotic scenarios, optimizing success rates or shortest-path achievement, relying on contrastive learning, behavior cloning regularization, or conservative value penalties, with **limited discussion of causal identifiability starting from observational distributions**. Our SCTA setting has three essential differences: (i) **latent confounding exists**, requiring proof under standard assumptions that soft Bellman expectations are identifiable from observational distributions (Proposition 4.1), rather than assuming simulability or interactivity; (ii) rewards are sparse signals of "first-hitting goal sets," and the task is not "predicting future observations" but pushing low-dimensional clinical outcomes toward the goal set $T$; (iii) offline $\pi\_b \to \pi\_\theta$ distribution shift is significant; we apply clipped ratios to immediate rewards to maintain operator contraction and provide interpretable bias bounds. For further comparison, we added the SCRL baseline (NCE-style contrastive critic, goal-conditioned actor with BC regularization, layer normalization, warm-start, tabular/temporal augmentation, and large-batch negative samples), but due to our task's "goal ≠ future observation" nature, contrastive learning primarily learns static similarity rather than temporal reachability, resulting in poor performance. In summary, our distinction lies in: centering on SCTA, we construct an integrated closed loop of theory-algorithm-evaluation around **causal identifiability and offline stability**, rather than pursuing general visual/manipulation achievement in confounding-free settings.

---

### Official Review · Reviewer_v1xV · 2025-10-31

**Soundness:** 2
**Presentation:** 3
**Contribution:** 2
**Rating:** 4
**Confidence:** 3

**Summary:**

The paper introduces GIFT, a goal-conditioned offline reinforcement learning framework for learning adaptive intervention policies from observational data. The task consists in selecting sequential interventions to drive a temporal system toward a desired target outcome, using only offline observational data. The method combines HER to label trajectories with new goals and reward rescaling using clipped importance weights. The method is shown to theoretically converge with bounded bias. Experiments on synthetic tumor growth and semi-synthetic MIMIC datasets are presented.

**Strengths:**

- the paper is well written and nice to read. It is overall clear and concepts are well explained.
- the problem setting of sequential counterfactual target achievement is a relevant task, that has also been explored in different fields like control, RL and causality
- the paper addresses two aspects met in sequential decision making of sparsity of success signals and distributional shift with HER and clipped importance weighting (similarly to offline RL regularization)
- the paper combines both theoretical results and experimental analysis, showing better performance in generalization and efficiency compared to the selected baselines

**Weaknesses:**

- since the motivation and potential impact of this work are strongly tied to real-world applications, the paper should more clearly discuss the practical validity of its assumptions. Causal-based assumptions are only briefly mentioned in the main paper but they are very strong assumptions that will not actually be met in most real-world scenarios.
- the task of target-driven policy optimization is a well known and studied problem in a variety of fields, ranging from treatment planning model-free/model-based RL, and model-predictive control. The main difference I see with this approach is that here the focus is more on the reconstruction of an underlying causal model and then using the supposedly 'true' counterfactual trajectories modeling for finding optimal policy. However, many other approaches focus on similar tasks by optimizing for performance of optimal policy reconstruction. Even though some of the theoretical results might be less strong since no causal model is assumed, the assumptions are also weaker and in real-world settings these will be very competitive baselines. Most of the baselines used in the paper do not learn reactive (closed-loop) policies but are counterfactual-based methods with similar underlying assumptions. Other standard offline RL algorithms beyond CQL, such as vanilla Q-learning, SAC, or DQN trained with the same reward formulation, etc. and different model-based control baselines (e.g., PILCO, PETS, or other MPC-style methods) that explicitly learn system dynamics for similar tasks are not considered.
- the importance-weighted reward scaling introduces bias that depends on the clipping thresholds: in theorem 4.2 the fixed-point solution deviates from the true value by $\sup|r(1-\bar\rho)|/(1-\gamma)$. Tuning these clipping bounds $\epsilon_1,\epsilon_2$ is seems very critical since too narrow clipping injects bias and too wide risks high variance


Minor:
- "an novel" in line 065
- "Offline Date Augmentation" instead of Offline Data Augmentation
- section Experimentx

**Questions:**

- can the authors elaborate on the causal assumptions for observational data they use in the paper and when these are realistic?
- how can you guarantee that the assumption of a bounded function class for the Q approximator holds in practice?

---

> ### Author Response · Authors · 2025-11-17
> **Part I of the response**
>
> **Regarding “the validity of the paper’s assumptions in real-world environments”:**
>
> Thank you very much for pointing this out. Our identifiability and convergence analyses rely on **standard identifiability assumptions** (consistency, sequential ignorability/no unmeasured confounding, and overlap/positivity) as well as goal-independent short-term dynamics (Assumption 4.1), without requiring reconstruction of a full structural equation model. We do not claim these assumptions hold strictly in all real-world settings; rather, we adopt them as minimal working assumptions to ensure the causal identifiability of the soft Bellman operator and to control instability from offline distribution shift during training via reward relabeling with importance-weight clipping.
>
> **Regarding “relationships and comparisons with broader baselines such as RL/MPC”:**
>
> We sincerely thank the reviewer for the feedback. We recognize that the current writing may lead readers to categorize our work as “general offline GCRL,” which is not our research objective. We also realize that many reviewers shared this confusion. Accordingly, we clarified this in the **Consolidated Response and Clarification**, explaining the differences between our work and “general offline GCRL.”
>
> We understand the reviewer’s interest in closed-loop policy baselines and, in the revision, we added **a general GCRL baseline—Stable Contrastive RL (SCRL) [1]**. We implemented and tuned it following the original method, including an NCE-style contrastive critic, a goal-conditioned actor with behavioral cloning regularization, layer normalization, cold start, data augmentations adapted to tabular/temporal inputs, and large-batch in-batch negatives. Results are reported in the main text (**Figure 3, Tables 1–2**): under our setting, SCRL performs suboptimally. The root cause is that SCRL treats “goal ≈ future observation” and relies on abundant image-level hard negatives and temporal contrastive signals, whereas the goals in SCTA are low-dimensional clinical targets T, where the key is reachability rather than similarity. Contrastive learning in this context tends to capture static similarity instead of the temporal reachability needed to attain the target set, making it hard to drive an effective closed-loop achievement policy.
>
> Regarding “standard offline RL algorithms”: our backbone builds on SAC, but direct training in the offline setting suffers from severe out-of-distribution estimation bias. To this end, we provided two controls: (i) removing reward relabeling (w/o RR), which can be viewed as an approximation to “naive offline SAC,” yields markedly higher variance and error (Table 3); (ii) replacing our relabeling with CQL (with CQL), a strong baseline representing “critic-side conservatism,” still underperforms GIFT, indicating that merely depressing OOD Q-values is insufficient to align the training signal for goal achievement.
>
> As for model-based control baselines such as MPC/PILCO/PETS, our comparative approach uses “learned world models (which in our case predict **counterfactual outcomes**, including RMSN/CRN/CT/ACTIN/VCIP) + planning,” and at inference we provide two “approximate MPC” variants: stepwise greedy and full-horizon optimization (Appendices G and F). Experiments show that such open-loop “model+planning” approaches are highly sensitive to model mismatch under our setting and are computationally expensive at inference; we also provide efficiency comparisons in Figure 5. Unlike classical MPC, our training data are observational and **confounded**; without addressing identifiability and the shift from $\pi_b$ to $\pi_\theta$, the world model will be systematically biased, amplifying planning errors.
>
> The key point of GIFT is not “whose controller is stronger,” but to demonstrate that “under **identifiability** constraints, learning an offline, goal-achieving closed-loop policy is feasible and stable.” Accordingly, we chose a set of strong “world model + planning” baselines (RMSN/CRN/CT/ACTIN/VCIP) and, in the revision, additionally included a general GCRL baseline (SCRL) to complement the closed-loop dimension. Overall, these comparisons support our central claim of stably learning deployable closed-loop policies from observational data.
>
> [1] Zheng et al., 2024. Stabilizing Contrastive RL: Techniques for Robotic Goal Reaching from Offline Data. In International Conference on Learning Representations (ICLR), 2024.

---

> ### Author Response · Authors · 2025-11-17
> **Part II of the response**
>
> **Regarding “bias introduced by importance clipping and whether threshold selection is crucial”:**
>
> Thank you very much for raising this. Theoretically, we provide a clear bias bound; clipping affects only the one-step reward, and the modified soft operator remains a $\gamma$-contraction, ensuring convergence and improving stability in offline training. Practically, **we systematically studied the impact of the clipping range on performance**: fixing $\epsilon_1=0.01$, we sweep the upper truncation $\epsilon_2\in \{0.01,0.05,0.1,0.5,1.0,2.0,10.0,20,100,200 \}$. On MIMIC and Tumor ($\kappa=3$) with $\tau\in\{1,2,4,6\}$, small upper truncation ($\epsilon_2\le 0.1$) increases RMSE due to truncation bias, while very large upper truncation ($\epsilon_2\ge 20/100$) degrades performance due to increased variance and instability. There exists **a broad stable region** $\epsilon_2\in[1,10]$, with $\epsilon_2=2.0$ near-optimal in most cases; as $\tau$ grows, the task becomes harder and more sensitive to $\epsilon_2$. These curves and discussions are included in **“Sensitivity and Ablations” (Sec. 5.4)**.
>
> **Regarding “how the bounded function class assumption for the Q-approximator holds in practice”:**
>
> To make the “bounded function class” assumption hold in practice, we impose constraints from three angles:
> - we apply a bounded mapping (sigmoid) to the **policy outputs** to ensure that the action inputs to the value function lie in a finite domain;
> - we use representation networks for state encoding with **Lipschitz/regularization** control to suppress numerical amplification and extrapolation drift;
> -  we align the evaluation and current policy distributions via one-step, bidirectionally clipped importance-based reward relabeling, which **mitigates variance** and out-of-distribution amplification caused by extreme weights.
>
> Meanwhile, sparse and **bounded rewards** (0 or -1), combined with a **discount factor** $\gamma<1$, provide an upper-scale reference for numerical magnitudes. Together, these mechanisms constrain the function’s amplitude at the levels of input domain, representation, and target signal.
>
> Finally, we corrected several typos in the paper. If there are any specific points you would like us to expand upon, we would be happy to continue improving the manuscript.

---

### Official Review · Reviewer_37Nt · 2025-10-31

**Soundness:** 2
**Presentation:** 2
**Contribution:** 2
**Rating:** 2
**Confidence:** 3

**Summary:**

In this work, the authors study how to guide temporal systems toward target outcomes. In particular, the goal-conditional adaptive strategy has been proposed via variance-controlled importance weights. Both theoretical and empirical studies have been performed to showcase the potential of the developed algorithms.

**Strengths:**

1. The paper is easy to follow
2. In this work, both theoretical studies and empirical studies have been provided.

**Weaknesses:**

1. This work is based on the offline RL settings, and the authors also mentioned the challenges in distributional shift. However, how the developed algorithm addresses the problem of distributional shift needs to be justified from a theoretical standpoint.
2. The regret bound is not provided for evaluating the effectiveness of the learned policy.
3. The motivation for using the state-encoder and LSTM is not clear.
4. As the authors are using domain data for experiments, could the authors highlight the benefits of the algorithms in the domain problem?

**Questions:**

N/A

---

> ### Author Response · Authors · 2025-11-18
> **Part I of the response**
>
> Thank you for the thoughtful feedback and detailed questions.
>
> **Regarding "the distributional shift issue":**
>
> The distributional shift in the offline setting is essentially a mismatch in occupancy measures: $d^{\pi\_\theta}(\psi,a)\neq d^{\pi\_b}(\psi,a)$. The soft evaluation objective fundamentally requires computing $T\_{\pi\_\theta}Q$ under the target policy distribution, while we only have static data generated by $\pi\_b$. To address this, we rewrite $\mathbb{E}\_{a'\sim\pi\_\theta}[\cdot]$ as $\mathbb{E}\_{a'\sim\pi\_b}[\rho\,\cdot]$ under the observational distribution using the one-step importance ratio $\rho(\psi',a')=\frac{\pi\_\theta(a'|\psi')}{\pi\_b(a'|\psi')}$ (**Proposition 4.1 of the revised manuscript**), and rescale immediate rewards using the clipped ratio $\bar\rho=\mathrm{clip}(\rho,\epsilon_1, \epsilon_2)$ as $\tilde r=\bar\rho\,r$, thereby inducing a modified soft Bellman operator $T\_{\tilde\pi\_\theta}$. This mechanism's direct connection to distributional shift manifests in three aspects: First, importance weighting by definition "reweights" data from the $\pi\_b$ distribution to evaluation quantities under the $\pi\_\theta$ distribution, directly addressing the distribution mismatch caused by $\pi\_\theta\neq\pi\_b$; Second, the clipping parameter $\epsilon$ compresses overly large $\rho$ values in low-support regions back to a controllable range, effectively imposing a "trust region" on OOD areas, preventing policy updates using extrapolated $Q$ values from further deviating from data support; Third, the bias-variance trade-off is theoretically quantifiable: $T\_{\tilde\pi\_\theta}$ remains a $\gamma$-contraction (Theorem 4.2) with an explicit performance gap bound
> $\|Q^{\pi\_\theta}-Q^{\tilde\pi\_\theta}\|\_\infty\le\frac{1}{1-\gamma}\sup\_{\psi,a}|r(\psi,a)(1-\bar\rho(\psi,a))|$ (Theorem 4.3). Since rewards in our task satisfy $|r|\le 1$ (first hit yields $0$, otherwise $-1$), this can be written as $\|Q^{\pi\_\theta}-Q^{\tilde\pi\_\theta}\|\_\infty\le\frac{1}{1-\gamma}\sup\_{\psi,a}|1-\bar\rho(\psi,a)|$, directly characterizing "distributional shift intensity" as the deviation of the ratio from $1$; when $\pi\_\theta$ is closer to $\pi\_b$ or $\epsilon$ is more conservative, this bias is smaller. In implementation, we explicitly fit $\pi\_b$ (maximum likelihood) to obtain robust ratio estimates and use $\epsilon$ to suppress high variance. Experimental evidence aligns with theory: under strong distributional shift settings with "train/test intervention policy mismatch" (Table 1), GIFT achieves the lowest terminal RMSE; **the sensitivity curve for $\epsilon$ (Figure 6) exhibits a stable "sweet spot,**" while removing reweighting (w/o RR) significantly increases variance and degrades performance, and while conservative Q-learning can suppress OOD actions, it is overall inferior to ratio reweighting (Table 3).
>
> **Regarding "the regret bound issue":**
>
> Classical regret bounds are typically established under settings where "a globally optimal policy exists and the objective is cumulative return maximization," measuring the cumulative gap from the optimal policy; this work addresses the specific task of "**goal achievement/terminal error minimization**" rather than general return maximization, making standard regret analysis inapplicable. Accordingly, we provide theoretical guarantees directly matched to the task: under goal-independent dynamics and standard causal identifiability conditions, $T\_{\pi\_\theta}$ can be unbiasedly identified through one-step weighting under the observational distribution (Proposition 4.1); $T\_{\tilde\pi\_\theta}$ with reward reweighting maintains $\gamma$-contraction and converges to a unique fixed point (Theorem 4.2); the systematic bias introduced by clipping has an explicit operator difference upper bound $\|Q^{\pi\_\theta}-Q^{\tilde\pi\_\theta}\|\_\infty\le\frac{1}{1-\gamma}\sup\_{\psi,a}|r(1-\bar\rho)|$ (Theorem 4.3). For evaluation, we adopt terminal RMSE consistent with the task definition rather than cumulative regret, and significantly outperform baselines across multiple datasets and time horizons (Tables 1 and 2).

---

> ### Author Response · Authors · 2025-11-18
> **Part II of the response**
>
> **Regarding "the motivation for state-encoder and LSTM":**
>
> Our state $S\_t$ consists of variable-length full history $\bar{H}\_t$ and goal $g$. In the presence of time-varying confounding and partial observability, using only current observations is insufficient to satisfy the Markov property; $\bar{H}\_t$ needs to be encoded as an approximate "sufficient statistic." Balancing data scale and offline stability, we adopt parameter-efficient LSTM to encode $\bar{H}\_t$ as $z\_{h,t}$, map goal $g$ through MLP to $z\_{g,t}$, then fuse them for sharing by $\pi\_\theta$ and $Q$. Compared to heavier Transformers, LSTM is more efficient in this offline scenario, better absorbing sparse success signals when combined with HER's goal relabeling; efficiency evaluation shows **competitive parameter count, MFLOPs, and training/inference latency**, with closed-loop policy inference requiring no online sequential search (Figure 5). It should be emphasized that encoder choice should be weighed and selected based on specific scenarios (such as sequence length, data scale, noise characteristics and temporal structure, interpretability/computational constraints, etc.), but this is not within the scope of this paper; our core mechanism **does not depend on a specific encoder**; as long as goal-conditioned representation, HER relabeling, and the $T\_{\tilde\pi\_\theta}$ training framework remain unchanged, theoretical and algorithmic properties hold.
>
> **Regarding "domain benefits":**
>
> In sequential interventions requiring "targeted outputs" such as personalized medicine, GIFT shifts from the open-loop paradigm of "learn world model then search for fixed plan" to "goal-conditioned closed-loop deployable policy." This brings three direct benefits: **Real-time adaptivity**: the policy can dynamically adjust to disease progression history and specific goals, avoiding the fragility of static plans to stochasticity and model errors (Figure 4); **Computational efficiency:** inference requires no costly sequential optimization, with significantly reduced latency matching clinical real-time requirements (Figure 5); **Robust generalization**: under policy shift and strong confounding conditions, GIFT achieves the lowest terminal RMSE in both oncology and MIMIC semi-synthetic environments, with more pronounced advantages on "unseen intervention policy" tests (Tables 1 and 2), demonstrating superior goal reachability and error control capabilities.

---

> ### Comment · Reviewer_37Nt · 2025-11-27
> **Thank you for your responses**
>
> I thank the authors for their responses,  and I am satisfied with most of the responses from the authors, except for the domain benefits. Some benefits can not be fully supported by the results provided by the authors. Specifically, I cannot draw a clear connection between the dynamically adjusted policy and the disease progression history, based on the results. In addition, the expected efficiency of the algorithm in the small sample regime is not showcased. Overall, I acknowledge that the paper has been improved in this revision, but there are still some concerns that are not fully resolved. Therefore, I would like to increase my score to 4, while I do not mind if the paper is finally accepted.

---

> > ### Author Response · Authors · 2025-11-28
> > **Clarification of the domain benefits**
> >
> > Thank you very much for your positive assessment of our previous response and for raising the score. We take your remaining concerns about “domain benefits” very seriously, specifically the connection between policy and disease-course history, and efficiency in the small-sample regime, and we provide further clarification below.
> >
> > Regarding the linkage between the policy and the disease-course history, the case study in Figure 4 offers intuitive evidence. This figure clearly contrasts the fundamental differences between open-loop baseline methods (such as RMSN) and our closed-loop method (GIFT) in handling the evolution of the disease course. As shown, the intervention plan generated by RMSN is based on its predicted future trajectory (the purple dashed line), and although this plan may appear capable of achieving the target in theory, it is a static plan that ignores the actually realized disease-course history during execution. Consequently, when the real disease trajectory (the purple solid line) deviates from the prediction due to environmental stochasticity or model misspecification, the model cannot exploit the most recent historical information to correct its actions, which ultimately leads to a large deviation from the target. In contrast, GIFT (the green solid line) exhibits **real-time adaptivity**: at every decision step, it re-evaluates and adjusts its actions based on the **real-time updated actual disease-course history**. This mechanism tightly couples the policy with the evolving disease history, enabling timely correction of execution-time deviations, robustly steering the patient toward the treatment goal and effectively avoiding the vulnerability of static planning to prediction errors.
> >
> > Regarding efficiency in the small-sample regime, we would like to clarify that in the medical decision-making context this primarily manifests as the challenge of “successful samples being scarce” (i.e., a sparse-reward problem). In clinical data, trajectories that perfectly achieve a specific treatment goal are often very rare, effectively forming a “few positive examples” scenario. Our ablation study (Table 3) shows that if we remove the hindsight experience replay (HER) mechanism specifically designed to address this issue, the model’s performance degrades significantly (for example, in the Tumor $\kappa=4$ setting, RMSE deteriorates from 0.14 to 0.40). This confirms that GIFT can efficiently extract learning signals from a large number of “failed” or “suboptimal” trajectories, and can thus still efficiently learn an optimal policy under conditions of extreme scarcity of successful samples.

---

### Official Review · Reviewer_Jjqv · 2025-11-02

**Soundness:** 3
**Presentation:** 3
**Contribution:** 2
**Rating:** 4
**Confidence:** 4

**Summary:**

This paper introduces a goal-conditioned reinforcement learning approach to develop counterfactual intervention policies in clinical settings. This work proposes GIFT, a framework within which intervention paradigms can move into more proactive, online settings rather than traditional counterfactual intervention strategies that plan offline. This proposed approach is shown to be more accurate than existing offline planning approaches in two clinically based synthetic domains.

**Strengths:**

The paper clearly formulates the MDP definition of the problem setting. The literature review of prior offline planning approaches to sequential counterfactual policy development is exhaustive, to my knowledge. This helps outline the presumed importance of the GIFT framework. I found that the methodology presented in Section 4 was reasonable but unfortunately lacked specificity and completeness (more in the next section). I especially appreciated the use of HER and the reward rescaling components. I feel that these help provide clear components by which the proposed GIFT framework rests on. Obviously, there is more to to GIFT than just these but I found that conceptually, they are what helped solidify the overall translation of standard offline planning to a more proactive policy, this is borne out in the ablation analysis provided in Section 5.

**Weaknesses:**

Despite my overall positive impressions of the proposed GIFT framework, I do not feel that the writing of the paper is appropriately detailed or scoped. I'm not entirely sure that the method is properly evaluated against relevant baselines given its construction and training. The paper, while originating in the counterfactual policy optimization line of literature, is more adequately viewed as a purely goal-conditioned RL work. There is not sufficient literature analysis in offline goal-conditioned RL (there has been major work coming out of Ben Eysenbach's lab along these lines the last two years) nor sufficient benchmarking. Granted the evaluations (accuracy of interventional outcome vs. policy return) of these two communities differ immensely but I feel that it's not appropriate to ignore one of them entirely. Because of this major omission, I cannot view the introduction of GIFT as wholly original or significant.

What could have changed my perspective of GIFT was a very detailed procedural overview of GIFT. There are a lot of details that are not provided about the step-by-step flow of information provided in a high-level fashion in Figure 2. I felt that the most important details missing were those about how the encoded state and goals came together, how they were used in the policy, etc and then finally what the intended output of the neural network was. I was surprised that the latter set of details were not outlined clearly. My guess is that the network provides both an estimate of the next observation as well as the clinical target measure $y_t$? Most egregious of these omitted details is an information about what the intended clinical measures that were used as targets. For the tumor task, it is pretty self-evident that this would be tumor size. But for the MIMIC task there were no details about what the intended clinical task was. After digging into the data details for the semi-synthetic use case of MIMIC data, my concerns deepened greatly. The patient trajectories are stated to be sampled randomly from the database. This is irresponsible and non-standard practice as there is a large spread of heterogeneous patient presentations with various clinical conditions they are being treated for within the MIMIC database. Even then, there is still zero indication what the designed target variables are. From this, I am very wary of the experimental design and have a hard time trusting the results.

There are several additional elements of the work that are very low in quality. The usage of $\tau$ as a temporal horizon (I'm guessing) is never articulated. This makes understanding Figures 3, 4, 6, and corresponding tables difficult to understand. A simple explanation about how predictions over extended horizons increase the difficulty and then clearly identify the notation used to represent this would go a long way. The same concerns arise in the use of $\gamma$ to identify different levels of confounding in the tumor dataset. First, the use of $\gamma$ is confusing since this is common used as a discount factor in the RL literature. So this introduces some confusion at the outset. Secondarily, the effect of confounding through the parameter is never described in order to delineate the importance of separate investigations in Figure 3 and on.

There are a lot of missing details about the datasets used to support the experimental analysis in Section 5. Yes, many of these are included in the appendix but there needs to be some detail to provide appropriate context for the main body of the paper. It's not clear what the "target trajectory" corresponds to in these offline settings. The goal relabeling HER procedure is also not sufficiently detailed. Was the goal set to just be the terminal state of the various trajectories? Was there any point where the training data was segmented such that a single trajectory could be bootstrapped using multiple goals at various temporal horizons?

In short, I think that the paper could be greatly improved with a far more careful re-write, where technical details about the proposed method are elevated and made central to the paper. Doing so while also working to more clearly differentiate the proposed methods from already established GCRL methods would help to strengthen the work. The clinical+interventional use case is well motivated and of high impact.

**Questions:**

Beyond those questions outlined in the above "Weaknesses" section, I have the following questions:

- In Section 4.3, the reward rescaling approach is not well motivated or justified. What is the purpose for the clipping mechanism? Why wasn't the interval for clipping parametrically studied? What is the IS ratio used for? Is it to counteract intrinsic biases that arise in the relabeling of the goals? More details about this approach would be appreciated. Also, in the IS ratio, is the behavior policy just the observed dynamics from the offline data?

- What is the RMSE calculated between? Is it averaged over the entirety of the trajectory? Or is it accumulated in other ways?

- Table 1's caption raises some confusion. Is the RMSE reported (and the results through Section 5) computed over both the training and test sets?

- At the end of Section 5.2 a separate experiment using the Tumor dataset is outlined where different intervention policies are used to generate target outcomes differing from the training data. Where are these policies derived from? What does this entail?

- What is the dotted purple line in Figure 4? It is missing from the legend.

---

> ### Author Response · Authors · 2025-11-17
> **Part I of the response**
>
> Thank you very much for your careful engagement with the paper and for your concern about its positioning and technical details. Your feedback is invaluable for improving our work. Below are our detailed responses and revisions:
>
> **Regarding the concern “this looks like pure GCRL and lacks related work and baselines”:**
>
> We sincerely thank the reviewer for the feedback. We recognize that the current writing may lead readers to categorize our work as “general offline GCRL,” which is not our research objective. We also note that several reviewers shared this confusion. Accordingly, in the **Consolidated Response and Clarification** we explicitly differentiated our work from “general offline GCRL,” and reported our reproduced and tuned **Stable Contrastive RL (SCRL) baseline [1]**. In brief, the fundamental difference between GIFT and common GCRL is that we operate in a causal–counterfactual–offline setting, where we must simultaneously ensure counterfactual identifiability and robust estimation under distribution shift; our rewards, operators, and theoretical guarantees are all built around these requirements. Related discussions and baseline results are presented in Sec. 2 & A.2 and 5.2.
>
> **Regarding “unclear procedural details”:**
>
> In the Methods section, we have completed the missing details regarding “how states are encoded, how goals are fused into the state, what inputs the policy receives, and what the network outputs.” Specifically, the historical trajectory is temporally encoded and then concatenated with the goal representation, **which is fed into a neural network for further fusion**. The policy network only outputs a continuous intervention vector $a_t\in(0,1)^{d_a}$; we do not output the next observation or an additional goal-prediction head. These details are clarified in **Sec. 4.1**.
>
> **Regarding “notation confusion”:**
>
> We consistently use $\tau$ to denote the evaluation horizon (rolling out at most $\tau$ steps; the task becomes harder as $\tau$ increases), and $\kappa$ to denote the confounding strength in the Tumor environment; the discount factor remains $\gamma$. All first occurrences and figure captions explicitly state these meanings to avoid conflicts with common RL symbols.
>
> **Regarding “data and clinical tasks”:**
>
> The MIMIC experiments are conducted in a controlled “semi-synthetic” environment: we construct a reproducible cohort using fixed inclusion criteria and random seeds, and then **synthesize continuous interventions and 2D clinical outcomes $(y_1,y_2)$ via a specified generative mechanism**. The task is to steer this 2D outcome into a predefined target region. In the Tumor task, the goal is to drive tumor volume into a target range. We explain this process and its rationale in **Sec. 5.1 and C.2** to avoid the impression of “unconstrained random sampling.”
>
> **Regarding “whether HER uses only endpoints and whether multi-goal segmentation is possible”:**
>
> We do not use only endpoints. For each transition, we sample **multiple achieved outcome points** from several future time steps as substitute goals, recompute the corresponding immediate rewards, and write them into the goal-conditioned replay buffer, thereby constructing multiple training goals from a single trajectory.
>
> **Regarding “reward relabeling and clipping” (Sec. 4.3):**
>
> We use the one-step importance ratio $\rho=\pi_\theta/\pi_b$ to reweight immediate rewards in order to align the evaluation operator with the current policy distribution under the offline setting while keeping contraction and providing a clear bias bound; clipping suppresses variance explosions from extreme ratios and controls bias. The behavior policy $\pi_b$ is not the “observation dynamics,” but a conditional density model we fit by maximum likelihood on offline data to estimate the denominator. Following your suggestion, **we systematically studied the effect of the clipping range on performance**: fixing $\epsilon_1=0.01$, we sweep the upper truncation $\epsilon_2\in\{0.01,0.05,0.1,0.5,1.0,2.0,10.0,20,100,200\}$. On MIMIC and Tumor ($\kappa=3$) with $\tau\in\{1,2,4,6\}$, we find that small upper truncation ($\epsilon_2\le 0.1$) raises RMSE due to truncation bias, whereas very large upper truncation ($\epsilon_2\ge 20/100$) degrades performance due to increased variance and instability. **There exists a broad stable region** $\epsilon_2\in[1,10]$, with $\epsilon_2=2.0$ near-optimal in most cases; as $\tau$ increases, the task becomes harder and more sensitive to $\epsilon_2$. The corresponding curves and discussion are included in “Sensitivity and Ablations” (Sec. 5.4).
>
> [1] Zheng et al., 2024. Stabilizing Contrastive RL: Techniques for Robotic Goal Reaching from Offline Data. In International Conference on Learning Representations (ICLR), 2024.

---

> ### Author Response · Authors · 2025-11-17
> **Part II of the response**
>
> **Regarding “evaluation protocol”:**
>
> RMSE is computed at the **endpoint on the test set**. Concretely, we roll out the policy for horizon $\tau$, take the terminal outcome $Y_{\text{term}}$, and compute the error against the $\tau$-step target $Y_{\text{target}}$; we do not average or accumulate over the entire trajectory. Unless otherwise stated, all figures and tables adopt this unified protocol. Table 1 and other tables/figures report **test-set results only** (mean ± standard deviation over multiple independent runs), with no mixing of training data.
>
> **Regarding the “unseen-policy” experiment (end of Sec. 5.2):**
>
> At test time, with probability $\eta$ we override the standard assignment and sample actions from a static $\mathrm{Beta}(\alpha,\beta)$ distribution, thereby intentionally introducing an intervention and target distribution different from training, to assess generalization to unseen intervention strategies. The specific settings and implementation details are provided in the main experiments and appendix.
>
> **Regarding “the purple dashed line and missing legend in Fig. 4”:**
>
> The purple dashed line corresponds to “RMSN **Prediction**,” while the others are **realized** trajectories; the target at $\tau=6$ is marked with a yellow star. We have completed the legend and caption and clarified this consistently in the text.
>
> Finally, following your suggestions, we have made concentrated revisions to the sections on related work; methods (state/goal encoding, HER, reward relabeling); experimental setup (notation, metrics, goal definition); main experiments; sensitivity analyses; and figure/table captions. We aim to improve clarity in positioning, detail, and readability. If there are any further points you would like us to elaborate on, we are happy to continue improving the manuscript.

---

> > ### Comment · Reviewer_Jjqv · 2025-11-24
> > **Thank you**
> >
> > Thanks for the detailed follow up to my review and several questions. The revision has significantly improved the paper. I am however concerned about two major aspects that have gone unaddressed and am uncertain that these can be resolved for this submission. These issues are central to the positioning of the paper and its perceived novelty. Without addressing these, I am unwilling to update my score. As mentioned, I am not sure that these can be done in the period allowed for consideration at ICLR this year.
> >
> > 1.) The categorization of GCRL as purely motivated by or developed for pixel-input robotic policies seems to avoid the point made by myself and the other reviewers. By proposing SCTA from a lens of goal conditioned MDPs, you are in fact doing goal conditioned RL. Granted to objective and constraints are different, as they should be, but it's disingenuous to conflate calls for better grounding with GCRL as not applicable.
> >
> > For future versions of this paper, I would recommend that the authors embrace that their approach is deploying GCRL albeit in a new setting and requiring specific adaptations to ensure its appropriateness. This would warrant a more thorough engagement with GCRL literature beyond just the few papers that have looked at offline settings of the approach. The novelty of the work is in clearly differentiating from the known GCRL settings that the intended problem necessitates. This latter expectation seems to be addressed by the authors where they have made the desiderata of their proposed framework more clearly articulated. However, it is necessary to recognize that even in new settings a solution approach does have a basis elsewhere and deserves to be acknowledged faithfully.
> >
> > 2.) I am dissatisfied by the response given by the authors regarding my questions about the MIMIC dataset used and the clinical objectives they sought to center their study on. My trust in these experiments and their validity for the work presented in this paper was negatively impacted by the vague response provided by the authors. I expected a clear and concise statement that would describe the cohort and clinical objective used to determine the exclusion criteria. I have little confidence that the proposed methods developed in this paper would be actually of use.

---

> ### Author Response · Authors · 2025-11-28
> **Response to the first concern**
>
> Thank you very much for your further comments on the positioning and novelty of our work. We realize that our previous response may have caused some misunderstanding. We would like to first clarify one point: from the very beginning, we have always understood this work as **an offline goal-conditioned RL (GCRL) method in the SCTA setting**, rather than trying to distance ourselves from the GCRL paradigm. When we previously emphasized in the response and in the main text that “our work does not focus on generic visual/robotic goal-reaching, but on SCTA with a causal interpretation,” our intention was to highlight the **specificity at the level of application and assumptions**, not to deny the applicability of GCRL. If our current wording gives the impression that “we believe GCRL is only suitable for pixel-based robotics and therefore irrelevant to SCTA,” that is due to our imprecise phrasing, and we will clarify this in the revision.
>
> More specifically, we fully agree with the reviewer’s point that, in terms of methodological form, GIFT is clearly **an RL algorithm on a goal-conditioned MDP**, and therefore falls within the scope of GCRL. Our contribution does not lie in “bypassing GCRL,” but rather in, on top of acknowledging this foundation, systematically introducing GCRL into a setting that has been relatively under-formalized so far—Sequential Counterfactual Target Achievement (SCTA)—and, within this setting, adding two aspects that are relatively underexplored in the existing GCRL literature: (1) **counterfactual identifiability** based on observed longitudinal data; and (2) **distribution shift handling and theoretical characterization** for transitioning from a behavior policy $\pi_b$ to an evaluation/target policy $\pi_\theta$ in the offline regime. To this end, we explicitly introduce standard causal assumptions such as consistency, sequential ignorability, and overlap, and on this basis we prove that, under the intervention mechanism $do(\pi_\theta)$, the soft Bellman operator can be identified from the observational distribution $P_{\text{obs}}$ via a single step of importance weighting (Proposition 4.1). At the same time, by rescaling the reward using truncated single-step importance weights, we construct a modified soft Bellman operator, prove that it remains a contraction in the $\ell_\infty$ norm (Theorem 4.2), and derive an explicit performance bound between its fixed point $Q^\omega$ and the true $Q^{\pi_\theta}$, where the upper bound exactly corresponds to the systematic bias introduced by truncation (Theorem 4.3). All of these analyses are carried out within a unified framework of “goal-conditioned RL + observational causal assumptions + offline $\pi_b \to \pi_\theta$.”
>
> We also accept the reviewer’s suggestion regarding the literature context. In the current version, the related work section, due to space limitations, uses visual/robotic tasks as the primary examples of GCRL applications, which indeed may be misread as us “equating GCRL with such applications.” In the revision, we will state more explicitly that these are only one important class of application scenarios for GCRL, rather than a characterization of the entire GCRL field. We have already discussed representative GCRL works such as C-Learning, Distributional Distance Classifiers, HIQL, and Stabilizing Contrastive RL in Appendix A.2, and we will move the parts most relevant to our paper into the main text. From an internal GCRL perspective, we will emphasize the contrast that existing methods mostly assume reward-free or unconfounded data and typically target success probability or shortest path, whereas SCTA in our setting requires learning a closed-loop policy on observational, potentially confounded clinical cohorts such that the counterfactual terminal outcome falls into a target set $T$, while simultaneously ensuring identifiability and offline stability.
>
> In summary, the contribution of this work is to demonstrate that **GCRL can be systematically extended and theorized in the new SCTA setting that combines observational data, causal identifiability, and offline distribution shift**. We are very grateful to the reviewer for pointing out the ambiguity and potential misleading aspects in our current positioning, which is extremely helpful for us to present the role of our work within the broader GCRL landscape more accurately.

---

> ### Author Response · Authors · 2025-11-28
> **Response to the second concern**
>
> We thank the reviewer for the careful examination of our data validity and clinical objectives. We have rewritten the relevant parts in the revised manuscript to more accurately reflect the actual processing pipeline used in our experiments:
>
> We would like to clarify that, in order to rigorously validate the effectiveness of the proposed algorithm, we require ground-truth counterfactual outcomes, which purely real-world clinical data cannot provide. Therefore, our experiments follow the semi-synthetic benchmark protocol established by Melnychuk et al. [1] (Appendix K). The experimental objective is not to directly model a single specific clinical condition (such as sepsis), but rather to control synthetic health states driven by realistic physiological dynamics. This setup leverages 25 vital signs from MIMIC-III (e.g., heart rate, blood glucose) to preserve the heterogeneity and complexity of real physiological data, while using synthetic outcomes (generated by Gaussian processes and spline functions) to provide the \emph{causal ground truth} that is not available in real-world datasets. It is worth noting that constructing semi-synthetic data based on real covariate distributions to evaluate counterfactual estimation algorithms is a widely adopted standard practice in the causal inference literature.
>
> Second, regarding the exclusion criteria, we removed all records with a length of stay shorter than 60 hours to ensure sufficient temporal history; we used fixed random seeds to sample $N=500$ independent patient trajectories; and we strictly truncated all trajectories to a fixed length of 60 hours, with missing values imputed using forward and backward filling. The resulting state space contains 25 real-valued dynamic vital signs (such as heart rate and blood glucose) and 3 static demographic features.
>
> [1] Valentyn Melnychuk, Dennis Frauen, and Stefan Feuerriegel. Causal transformer for estimating counterfactual outcomes. In International Conference on Machine Learning, pp. 15293–15329. PMLR, 2022.

---

### Author Response · Authors · 2025-11-17
**Consolidated Response and Clarification**

We sincerely thank the reviewers for their valuable comments. We recognize that the current exposition may lead readers to categorize our work as “general offline GCRL,” which is not our research objective. To eliminate this potential misunderstanding, we provide a unified response and clarification here.

It is important to clarify that our object of study is **Sequential Causal Counterfactual Target Achievement (SCTA)**, whose core aim is to learn, from observational data, a clinically/population-oriented target achievement policy, rather than a general visual/robotic goal-reaching task.

Our core contribution is not “designing a GCRL algorithm,” but demonstrating that formulating SCTA as a goal-conditioned MDP and combining it with stable training on offline data is both feasible and theoretically guaranteed. Specifically, SCTA must handle, in the offline setting, (1) **causal identifiability (explicitly articulated in Proposition 4.1 of the revised manuscript)** and (2) mismatch between the behavior ($\pi\_b$) and evaluation ($\pi\_theta$) policies (distribution shift). GIFT unifies these two aspects into a convergent Bellman framework via goal conditioning + HER + reward relabeling with bounded importance weights; we have clarified this more explicitly in the revision.

Compared to GCRL methods primarily evaluated in robotic vision settings (e.g., Stable Contrastive RL), SCTA aims to learn a closed-loop policy that drives counterfactual outcomes into a target set $T$ from confounded observational data. This setting differs from visual goal reaching and requires explicit attention to counterfactual identifiability and robust estimation under offline ($\pi\_b \to \pi\_\theta$) distribution shift.

For a more direct comparison, **we implemented and tuned the Stable Contrastive RL baseline** with the following core components: an NCE-style contrastive critic (scoring via the inner product between $\phi(s,a)$ and $\psi(g)$, see Eq. 2 in the original paper), a goal-conditioned actor with a behavioral cloning regularizer (Eq. 4), layer normalization, cold initialization (±$1e^{-12}$), data augmentation (adapted to tabular/temporal inputs), and large-batch in-batch negatives. **Implementation details are provided in Appendix G.1**.

SCRL’s empirical results are **suboptimal (Fig. 3, Tables 1 and 2)**: under on-policy train/test in the MIMIC semi-synthetic setting (Table 2), at $\tau=6$ SCRL achieves $0.65\pm0.30$, substantially worse than GIFT’s $0.39\pm0.18$; under unseen-policy generalization in Tumor ($\kappa=4$, Table 1), the gap is larger—at $\tau=6$ SCRL reaches $2.42\pm0.38$ whereas GIFT is only $0.46\pm0.08$, and across all $\tau\in\{1,\dots,6\}$ on this dataset, SCRL is consistently inferior to GIFT. Figure 3 further shows that as confounding strength increases from $\kappa=2\to4$ and $\tau$ grows, SCRL’s curves lie above GIFT and degrade more rapidly. A plausible reason is that SCRL assumes “goal ≈ future observation” and relies on abundant image-level hard negatives and temporal contrastive signals; in our setting, goals are low-dimensional clinical targets rather than “future observations,” making contrastive learning prone to capturing static similarity instead of the temporal reachability required to attain targets. Moreover, it does not explicitly address “observational confounding + offline ($\pi\_b \to \pi\_\theta$) distribution shift,” impeding stable and effective target achievement.

Reference:
Zheng et al., 2024. Stabilizing Contrastive RL: Techniques for Robotic Goal Reaching from Offline Data. In International Conference on Learning Representations (ICLR), 2024.

---

### Meta-Review · Area_Chair_A4ut · 2026-01-08

**Summary:**

This paper proposes GIFT, an offline goal-conditioned RL framework for sequential counterfactual target achievement, built by combining hindsight experience replay with clipped importance-weighted reward rescaling and a soft-Q convergence/bias analysis. While the problem setting is important and the paper is clearly written, I agree with Reviewers Jjqv and equA that the core methodological contribution is incremental. HER and clipped importance weighting are standard tools in the RL literature, and the theoretical results largely follow routine contraction-mapping and bias-bound analyses. The work does not convincingly differentiate itself from existing offline goal-conditioned RL or conservative policy learning methods, and the added SCRL/CQL comparisons are helpful but do not change that basic picture.

On the empirical side, the experiments are thorough but not fully convincing as evidence of a substantial new idea. Reviewer Jjqv raises reasonable concerns about the construction and interpretation of the semi-synthetic MIMIC environment and the lack of a clear, clinically meaningful target task; despite a detailed rebuttal, they explicitly state that their trust in the practical relevance of the results remains low. Reviewer equA similarly questions the realism of the causal assumptions and the absence of stronger baselines from standard offline RL and model-based control. Although the rebuttal improves clarity (e.g., better description of state/goal encoding, notation, clipping sensitivity, and dataset details), it does not resolve these core issues around novelty, positioning, and real-world validity. I therefore recommend rejection.

**Reviewer Concerns:**

Reviewer Jjqv. The rebuttal adequately clarified several concrete issues: it made explicit that the setting is offline goal-conditioned RL for SCTA, added a tuned SCRL baseline, and gave more detail on the network inputs/outputs, goal encoding, and the construction of the semi-synthetic MIMIC environment. However, Jjqv’s deeper concerns remain: the clinical targets and semi-synthetic design still feel underspecified and somewhat ad hoc, and the work is not convincingly positioned relative to the broader offline GCRL / policy-optimization literature, so questions about originality and practical trustworthiness of the results are largely unresolved.

Reviewer 37Nt. The rebuttal addressed some of 37Nt’s technical questions by explaining how the clipped importance weights are intended to control distribution shift and by strengthening the convergence discussion (Proposition 4.1). That said, the reviewer’s core concerns about theoretical treatment of offline distribution shift and the absence of any regret or performance guarantees remain essentially open, and the motivation for specific architectural choices (state encoder, LSTM) is still only loosely justified.

Reviewer v1xV. For v1xV, the authors improved the exposition of assumptions and added discussion of related work, and they argued why some standard offline RL baselines are less aligned with their SCTA goal. Nonetheless, the key points that the causal assumptions are strong and only lightly justified, that important reactive RL / model-based control baselines are missing, and that the bias introduced by importance-weighted reward scaling is not carefully analyzed, are not fully resolved.

Reviewer equA. The unified response directly engages with equA’s comments by clarifying that GIFT is best viewed as an offline GCRL method specialized to SCTA, and by expanding comparisons to SCRL and CQL. Still, equA’s main critique—that the method is largely a recombination of HER and clipped importance weighting with standard contraction/bias analysis, and that the goal-conditioned framing of counterfactual control is not a major conceptual advance over existing work—remains valid in my view; the rebuttal does not turn this into a clearly novel or theoretically compelling contribution.

**Reviewer Scores:**

The reviewers seemed to be unsatisfied with some of the authors' responses. I guess they may keep their original score.

---

### Decision · Program_Chairs · 2026-01-26

Reject